# Search performance and octopamine neuronal signaling mediate parasitoid induced changes in *Drosophila* oviposition behavior

Lan Pang [1,2,3], Zhiguo Liu [1,2,3], Jiani Chen [1,2,3], Zhi Dong [1,2,3], Sicong Zhou [1,2,3], Qichao Zhang [1,2,3], Yueqi Lu[1,2,3], Yifeng Sheng [1,2,3], Xuexin Chen [1,2,3,4] & Jianhua Huang [1,2,3] ✉

Making the appropriate responses to predation risk is essential for the survival of an organism; however, the underlying mechanisms are still largely unknown. Here, we find that *Drosophila* has evolved an adaptive strategy to manage the threat from its parasitoid wasp by manipulating the oviposition behavior. Through perception of the differences in host search performance of wasps, *Drosophila* is able to recognize younger wasps as a higher level of threat and consequently depress the oviposition. We further show that this antiparasitoid behavior is mediated by the regulation of the expression of *Tdc2* and *Tβh* in the ventral nerve cord via LC4 visual projection neurons, which in turn leads to the dramatic reduction in octopamine and the resulting dysfunction of mature follicle trimming and rupture. Our study uncovers a detailed mechanism underlying the defensive behavior in insects that may advance our understanding of predator avoidance in animals.

In nature, animals are exposed to a broad array of dangers, including threats from predators. The ability to detect and respond to predators is an innate behavior fundamental to survival that is conserved across species[1–5]. Studies of vertebrates and invertebrates have indicated that multiple sensory systems, including olfaction, audition, and vision, are involved in the sensing of predators. Furthermore, prey animals have evolved a variety of antipredator adaptations to avoid capture and to actively defend against predators[6–9]. For instance, rodent preys exhibit defensive behaviors with either escape or freezing patterns in response to carnivores[6,7]; Aegean wall lizards, *Podarcis erhardii*, generally present tail autotomy defenses against predation[9]. Although beneficial when predation risk is high, defensive responses come at a cost and, therefore, could potentially be inactive in low-risk environments[10–12]. For example, many eared insects can assess the level of threat in the different stages of a bat attack and decide whether to initiate defensive evasion since the response requires energy expenditure and diverts effort and time from other crucial activities

such as eating and mating[10]. Another example is that elk can eliminate a defensive strategy affecting reproductive physiology when the predation risk of wolves is relatively low[12]. Despite the characterization of multiple fascinating behavioral responses[13–17], a lack of empirical studies has limited our understanding of how prey animals precisely adjust antipredator behaviors depending on predation risk levels. In addition, the neuronal mechanisms underlying defensive responses remain poorly understood.

Parasitoid wasps are insects whose progeny develop by consuming and eventually killing their host species, which are generally other insects[18]. Thus, parasitoid wasps act as keystone species in maintaining natural ecosystems, and they are deadly natural enemies representing a special kind of predators who usually present symbiotic relationship with their prey animals (known as hosts) in the wild[19,20]. Similar to many insects, *Drosophila melanogaster* is regularly attacked by parasitoid wasps, with infection rates as high as 90% in natural populations[21,22]. When encountering cosmopolitan *Leptopilina* wasps, such as the

[1]Institute of Insect Sciences, College of Agriculture and Biotechnology, Zhejiang University, Hangzhou 310058, China. [2]Ministry of Agriculture Key Lab of Molecular Biology of Crop Pathogens and Insect Pests, Zhejiang University, Hangzhou 310058, China. [3]Key Laboratory of Biology of Crop Pathogens and Insects of Zhejiang Province, Zhejiang University, Hangzhou 310058, China. [4]State Key Lab of Rice Biology, Zhejiang University, Hangzhou 310058, China. ✉e-mail: jhhuang@zju.edu.cn

specialist *L. boulardi* (Lb) and the generalist *L. heterotoma* (Lh), adult *Drosophila* females undergo intriguing behavioral changes, including altered food preferences, reduced oviposition rates and accelerated mating behavior[23-25]. These defensive behaviors allow flies to protect their offspring because adult flies are not infected by *Leptopilina* female wasps, which lay their eggs inside the body of *Drosophila* larvae. Recently, it was discovered that threat information can be transmitted to naive adult *Drosophila* females that have never seen wasps before, eliciting similar defensive behaviors[26,27]. These studies suggest that *Drosophila* have developed to recognize and respond to parasitoid threats, most importantly through the visual system. However, it is still unclear whether they can discriminate the levels of wasp danger to reduce or even suspend defensive responses when predation risk is low, which is an adaptive strategy to balance the costs and benefits of antipredator behaviors.

To better understand how organisms utilize multisensory inputs to induce accurate responses to predators, we designed a long-term exposure experiment to detect oviposition changes in *Drosophila* females exposed to Lb. Strikingly, we found that *Drosophila* have distinct egg-laying rates when exposed to young and old Lb females. We showed that visual inputs result in depression of egg laying and that the level of depression differs depending on whether the wasps are young or old. We also systematically analyzed the underlying mechanisms and found that the reduction in octopamine (OA) neuronal signaling led to decreased oviposition.

## Results

### Diverse oviposition rates of *Drosophila* females after long exposure to wasps

To investigate whether *D. melanogaster* change oviposition behavior when they cohabit with Lb female wasps, we designed an experimental procedure and monitored egg laying for a much longer time than in previous experiments – approximately 20 days. Specifically, twenty 3-day-old female and five 3-day-old male *D. melanogaster* adults were placed in standard fly bottles containing fly food dishes. Flies were housed with twenty 2-day-old Lb female wasps (exposed) or without any female wasps (unexposed). The fly food dishes were replaced daily, and fly eggs were counted daily (Fig. 1a). Consistent with previous observations[24], the exposed *Drosophila* females had significantly reduced oviposition numbers compared to the unexposed flies (Fig. 1b). This response lasted approximately 6 days in the presence of Lb females. After that, we surprisingly found that the number of eggs laid by the exposed flies did not differ from the numbers laid by the unexposed controls (Fig. 1b). This variation led us to speculate that this decreased oviposition may have been induced by the diverse life-threatening pressure when *D. melanogaster* females encounter different aged wasps, as old ones present less danger to their offspring[28,29], or simply indicate that the flies become habituated to the constant presence of wasps.

To differentiate the two hypotheses, we performed another oviposition behavioral assay by adding the young and old wasps, respectively. Based on their effects on *Drosophila* egg-laying performance (Fig. 1b), the Lb females ranging from newly eclosed to 8 days old were considered as young wasps, while Lb females older than 8 days were considered as old wasps. In this assay, both flies and wasps were anesthetized by ice for a short time, and then, the Lb females were replaced by young (4-day-old) or old (12-day-old) female wasps on Day 7. The oviposition numbers were monitored for another 7 days. As expected, the exposed *Drosophila* females reduced their oviposition rate during approximately the first 6 or 7 days in the presence of female wasps (Fig. 1c, d). Interestingly, the *Drosophila* females in groups re-exposed to young wasps after Day 7 significantly responded with a reduction in the oviposition rate that lasted an extra 5 days (Fig. 1c). When flies were re-exposed to 12-day-old wasps, oviposition remained equivalent to that of the unexposed flies (Fig. 1d). Thus,

these results indicate that *D. melanogaster* females are able to distinguish young and old wasps and reduce oviposition only in the presence of young parasitoids but not due to habituation.

To further check whether the housing experience with Lb females might contribute to the flies' oviposition behavior, the 3-day-old naive flies were independently housed with different aged Lb females (Supplementary Fig. 1a). As expected, we found that the female flies, without any prior experience with Lb females, significantly decreased egg laying when they were exposed to young but not old wasps (Supplementary Fig. 1b). Moreover, the levels of oviposition reduction were much similar to flies that were continuously exposed to young Lb wasps (Supplementary Fig. 1c). These results suggest that the reduced oviposition behavior of exposed flies certainly requires no learning experiences.

### *Drosophila* reduces egg laying in the presence of young parasitoids

We next investigated whether exposure to other species of *Drosophila* wasps at a young age also impaired the egg-laying rate. We first tested Lh, which is closely related to Lb, and subsequently *Asobara japonica* (Aj); all of these species have been shown previously to attack *D. melanogaster* 2nd instar larvae. Similar to previous processes, 2-day-old female wasps of Lh or Aj were placed in fly bottles with twenty 3-day-old females and five 3-day-old male *D. melanogaster* adults, and the numbers of *Drosophila* eggs laid were continuously monitored for 12 days. We found that the fly females laid fewer eggs during approximately the first 8 days in the presence of Lh or Aj female wasps, and the effects were similar to those following exposure to young female Lb (Supplementary Fig. 2a, b). However, when flies were exposed to old Lh or Aj wasps, oviposition numbers were equivalent to that of the unexposed flies. As a further test, when Lb males or an extra number of *D. melanogaster* males were present, the female flies displayed no such decrease in oviposition behavior (Supplementary Fig. 2c, d). To identify whether the presence of any other insects whatsoever reduces *D. melanogaster* egg laying, we used another species of *Drosophila*, *D. suzukii*, and the other two nonpredatory parasitoid wasps to *D. melanogaster*, *Chouioia cunea* and *Scleroderma guani*. We found that the presence of *D. suzukii* males did not have any effects on the egg laying of *D. melanogaster* females, nor did the presence of *C. cunea* or *S. guani* young females (Supplementary Fig. 2e–g). As such, these results suggest that *D. melanogaster* has specifically evolved to distinguish female from male Lb and parasitic parasitoids from nonparasitic insects.

### Decreased oviposition is relevant to the potential parasitic efficiency

Focusing on the main finding of this study that the presence of young female parasitic wasps triggered decreased oviposition, we further investigated the difference between young and old wasp females. Some studies have already revealed that the parasitic efficiency of female wasps declines as they age[28,29]. We then performed an assay to detect whether parasitic efficiency was impaired in the young and old Lb female wasps in this system. We found that the parasitic rates were 92%, 88%, 78% and 67% for the 2-day-old, 4-day-old, 6-day-old, and 8-day-old young wasps, respectively, whereas the parasitic rate of old wasps (10-day-old and 12-day-old) dramatically decreased to 57% and 43%, respectively (Fig. 2a). To further identify the distinct life-threatening stimuli from the cohabiting Lb females, we generated two olfaction-defective wasp strains, including a strain with knockdown of *Orco* (a gene encoding an obligate coreceptor of all odorant receptor proteins) mediated by RNA interference (Fig. 2b) and an antenna-ablated strain. As expected, the *Orco RNAi*-treated and antenna-ablated female wasps presented low or zero parasitic ability (Fig. 2c), which is consistent with the fact that olfaction is important and necessary for host seeking of parasitoid wasps[30-32]. We then placed

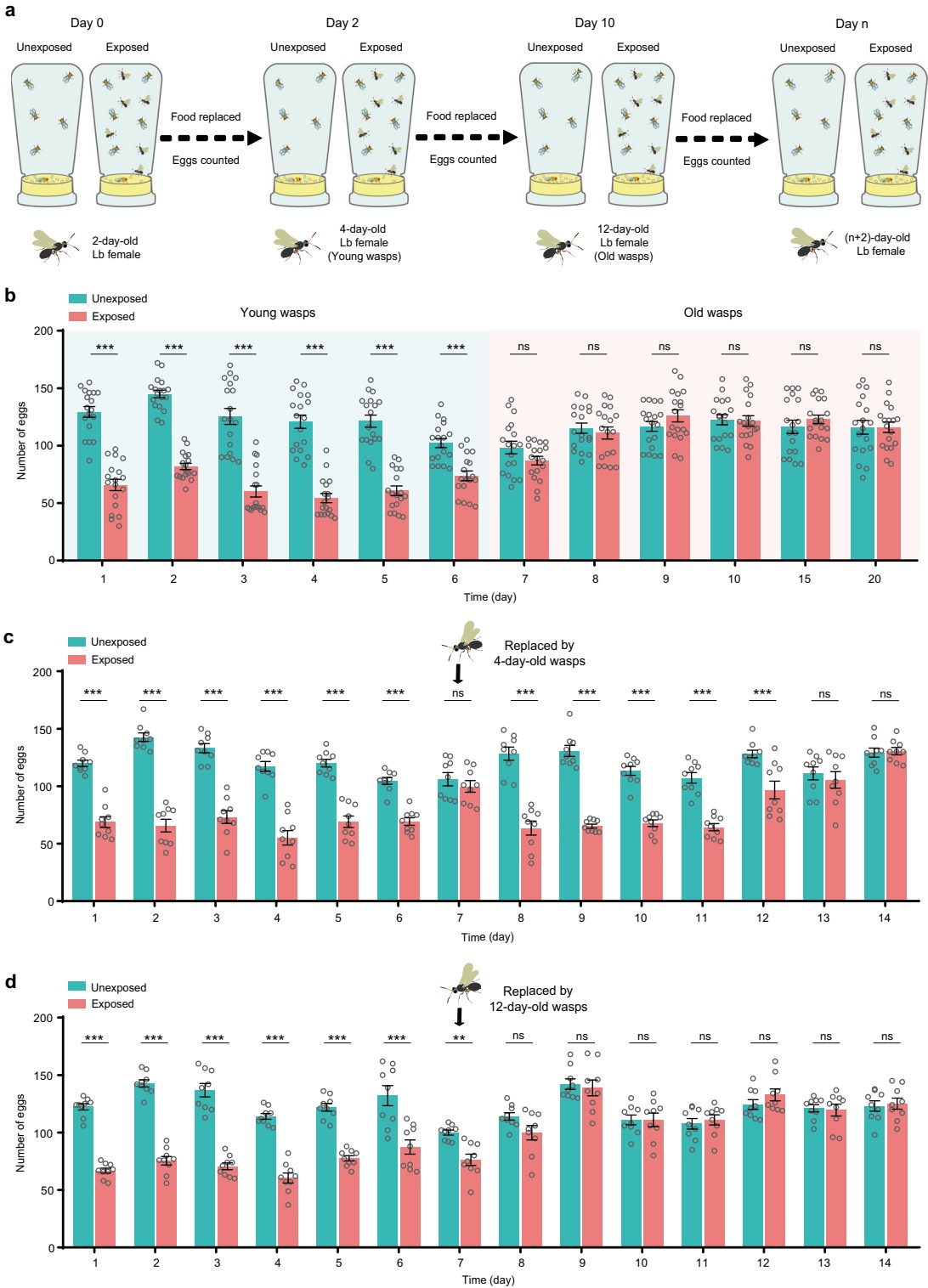

*Drosophila* females with these parasitism-defective 4-day-old young female wasps and detected the egg-laying behavior. We found that decreased oviposition was not observed upon exposure to *Orco RNAi*-treated female wasps and antenna-ablated female wasps (Fig. 2d). Overall, these results suggest that wasp parasitism behaviors might cause the decreased oviposition of flies exposed to young parasitoids.

## *Drosophila* oviposition depression is correlated with the host search performance of female wasps

We next sought to determine how the female wasps provided cues that elicit the changes in *Drosophila* oviposition because it is not possible for flies to directly obtain the parasitic rate of Lb. Once the Lb females were released into the fly food bottles, they began to search for hosts to parasitize on the food surface, and they also intermittently rested on the

**Fig. 1 | *D. melanogaster* oviposition rates are altered in the presence of young Lb females. a** Standard oviposition assay design. Each bottle contained twenty *Canton-S* (CS) female flies and five CS male flies, either with twenty female Lb wasps (exposed) or with no wasps (unexposed). Flies aged 3 days post-eclosion and wasps aged 2 days post-emergence were used. The food dishes were replaced daily, and the eggs laid each day were counted. **b** The daily number of eggs laid by the unexposed and exposed CS flies. Flies were exposed to wasps for 20 days. The experiment was performed eighteen times. Data represent the mean ± SEM. Significance was determined by two-way ANOVA with Sidak's multiple comparisons test, *p* values are indicated in Source Data file (***$p < 0.001$; ns, not significant). **c** The daily number of eggs laid by the unexposed flies and flies in bottles with

2-day-old female wasps at the beginning that were replaced by 4-day-old females on Day 7 (exposed). The experiment was performed nine times. Data represent the mean ± SEM. Significance was determined by two-way ANOVA with Sidak's multiple comparisons test, *p* values are indicated in Source Data file (***$p < 0.001$; ns, not significant). **d** The daily number of eggs laid by the unexposed flies and flies in bottles with 2-day-old female wasps at the beginning that were replaced by 12-day-old females on Day 7 (exposed). The experiment was performed nine times. Data represent the mean ± SEM. Significance was determined by two-way ANOVA with Sidak's multiple comparisons test, *p* values are indicated in Source Data file (**$p < 0.01$; ***$p < 0.001$; ns, not significant). Source data are provided as a Source Data file.

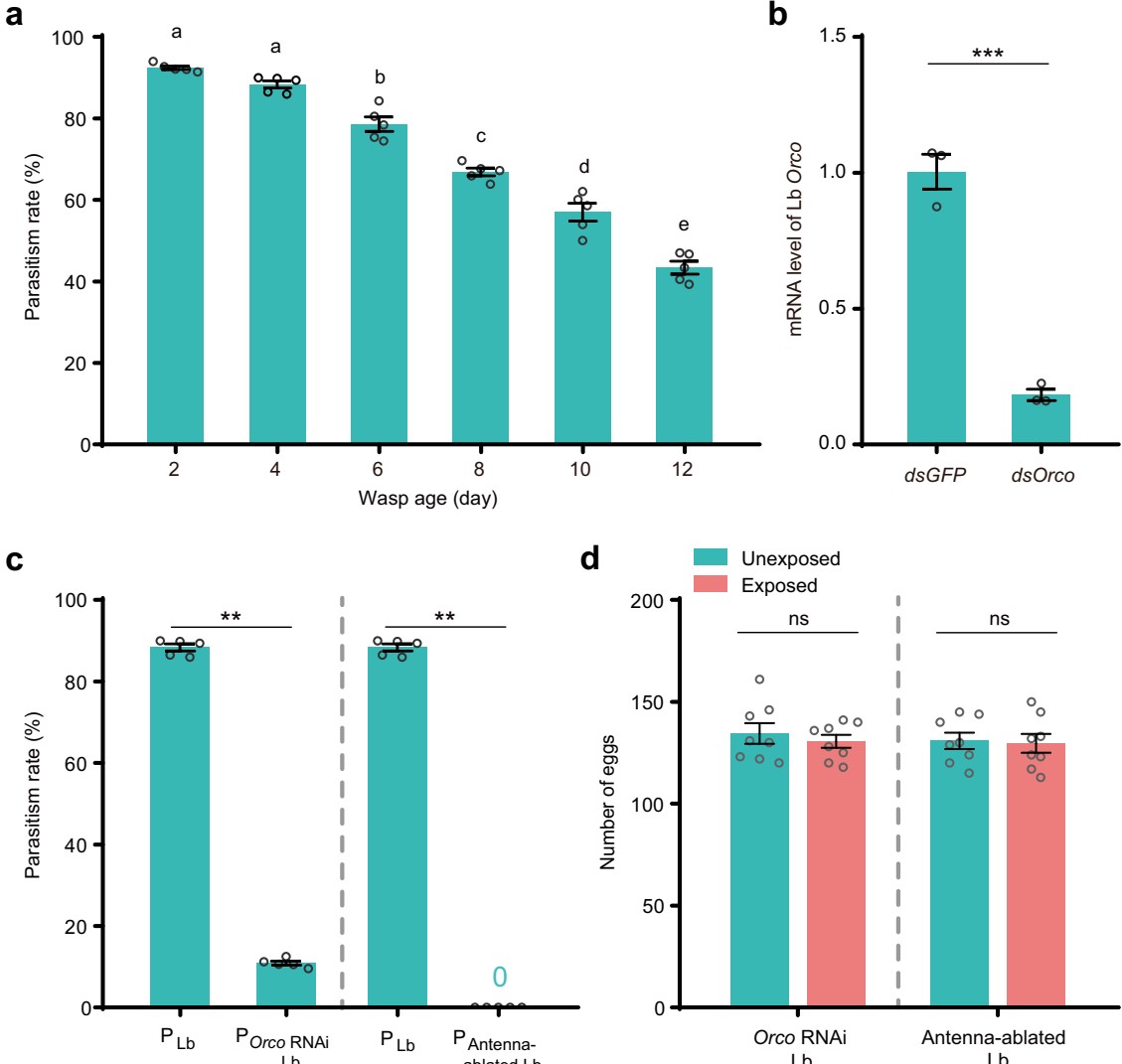

**Fig. 2 | Decreased oviposition is related to wasp parasitic efficiency. a** Parasitism rates in *D. melanogaster* host larvae attacked by Lb wasp females of the indicated ages. Left to right, *n* = 1062, 1139, 1351, 1130, 1055, and 1104 biologically independent host larvae. The experiment was performed five times. Data represent the mean ± SEM. Significance was determined by one-way ANOVA with Sidak's multiple comparisons test, *p* values are indicated in Source Data file. Different letters indicate statistically significant differences ($p < 0.05$). **b** Relative mRNA levels of *Orco* in Lb after RNAi treatment. *n* = 3 per group. Data represent the mean ± SEM. Significance was determined by two-sided unpaired Student's t test, *p* value is indicated in Source Data file (***$p < 0.001$). **c** Parasitism rates in *D. melanogaster* host larvae attacked by the *Orco* RNAi-treated ($P_{Orco\ RNAi\ Lb}$) and antenna-ablated

($P_{Antenna-ablated\ Lb}$) female wasps compared with the control ($P_{Lb}$) (attacked by 4-day-old female wasps in a). Left to right, *n* = 1139, 950, 1139, and 1196 biologically independent host larvae. The experiment was performed five times. Data represent the mean ± SEM. Significance was determined by two-sided Mann–Whitney U test, *p* values are indicated in Source Data file (**$p < 0.01$). **d** The daily number of eggs laid by the unexposed and exposed CS flies. Flies were exposed to the *Orco* RNAi-treated and antenna-ablated Lb female wasps. Egg numbers were counted on Day 2. The experiment was performed eight times. Data represent the mean ± SEM. Significance was determined by two-sided unpaired Student's t test, *p* values are indicated in Source Data file (ns, not significant). Source data are provided as a Source Data file.

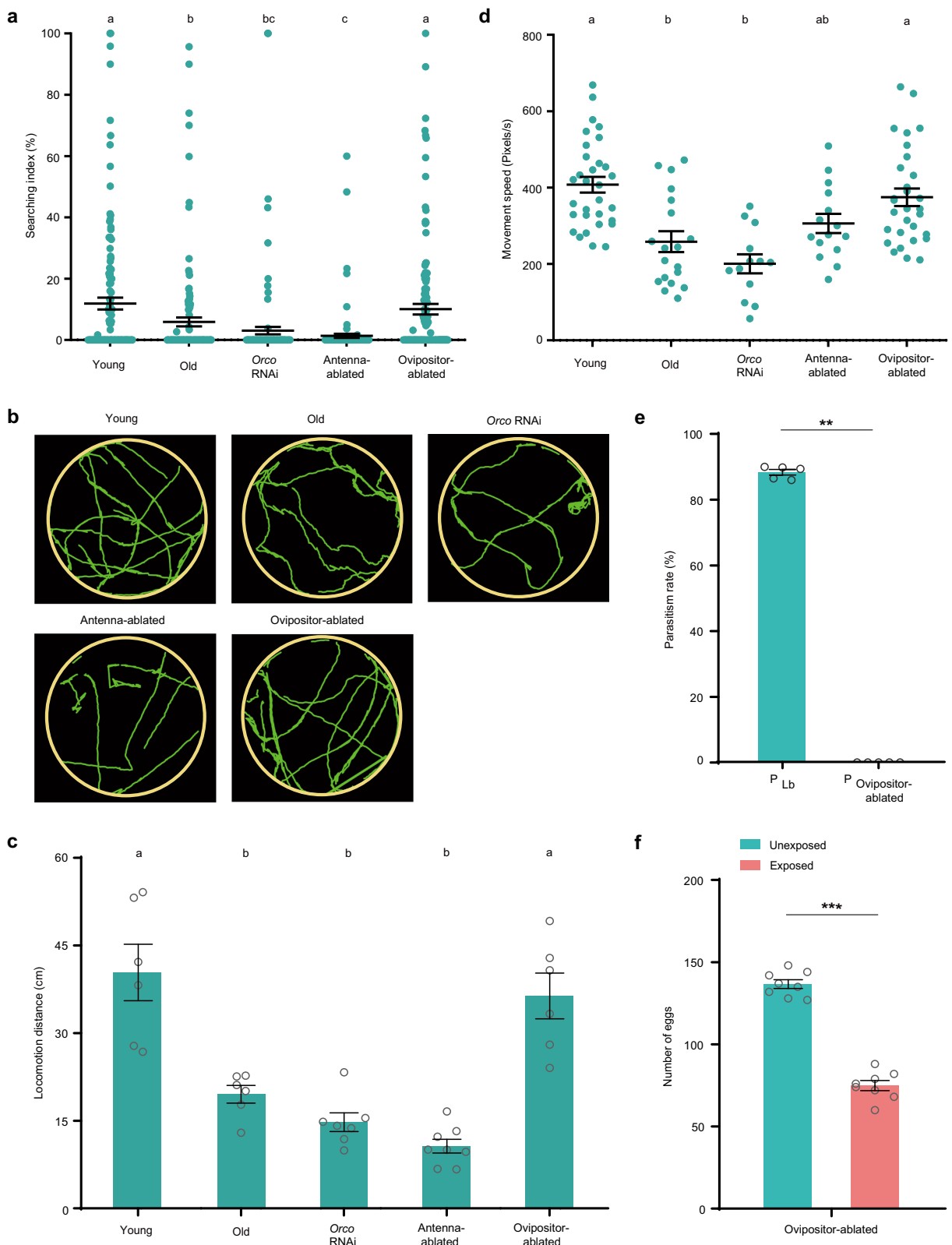

fly food or the fly bottle walls. The stereotyped search behavior of Lb female wasps includes rhythmically drumming with their antennal tips, continuous movement on food substrate and frequent stinging with their sharp ovipositor in fly food (Supplementary Movie 1). We then compared the search index (SI), which is defined as the percent of time a wasp presents search behavior in a certain period of observation time (10 min in this study). The results showed that the SI was significantly higher for 4-day-old than 12-day-old female wasps, although it was variable among different tested individuals (Fig. 3a). Correspondingly, locomotion trajectory analysis showed that the young females moved more on the food medium than did the old wasps (Fig. 3b, c), indicating that the old wasps spent much more time resting. Moreover, the resulting locomotion speed of the young female wasps was higher than that of the old wasps (Fig. 3d). To further investigate whether the

**Fig. 3 | Host search performance of Lb wasps is responsible for decreased *Drosophila* oviposition.** **a** The search index (SI) values (time spent searching in 10 min × 100) of different types of wasps. $n = 128$ for each group. Data represent the mean ± SEM. Significance was determined by Kruskal–Wallis test with Dunn's multiple comparisons test, *p* values are indicated in Source Data file. Different letters indicate statistically significant differences ($p < 0.05$). **b** Representative images of locomotion trajectories (green) of the different types of wasps in 2 min. There were 8 wasps for each record. At least six biologically independent experiments were performed. **c** Locomotion distance of the different types of wasps. Each plot indicates the locomotion distance in 3b (Left to right, $n = 6, 6, 7, 8,$ and 6 per type). Data represent the mean ± SEM. Significance was determined by one-way ANOVA with Sidak's multiple comparisons test, *p* values are indicated in Source Data file. Different letters indicate statistically significant differences ($p < 0.05$). **d** The locomotion speed of the different types of wasps. Left to right, $n = 31, 19, 13, 15,$ and 30 biologically independent Lb wasps. Data represent the mean ± SEM. Significance was determined by one-way ANOVA with Sidak's multiple comparisons

test, *p* values are indicated in Source Data file. Different letters indicate statistically significant differences ($p < 0.05$). **e** Parasitism rates in *D. melanogaster* host larvae attacked by the young ($P_{Lb}$) (attacked by 4-day-old female wasps in 2a) and ovipositor-ablated ($P_{Ovipositor-ablated}$) Lb female wasps. Left to right, $n = 1139$ and 972 biologically independent host larvae. The experiment was performed five times. Data represent the mean ± SEM. Significance was determined by two-sided Mann–Whitney U test, *p* value is indicated in Source Data file (**$p < 0.01$). **f** Egg numbers of CS flies exposed to ovipositor-ablated Lb female wasps compared to unexposed flies. Eggs were counted on Day 2. The experiment was performed eight times. Data represent the mean ± SEM. Significance was determined by two-sided unpaired Student's *t* test, *p* value is indicated in Source Data file (***$p < 0.001$). Wasp types: Young, 4-day-old Lb female wasps; Old, 12-day-old Lb female wasps; *Orco* RNAi, 4-day-old *Orco* RNAi-treated Lb female wasps; Antenna-ablated, 4-day-old antenna-ablated Lb female wasps; and Ovipositor-ablated, 4-day-old ovipositor-ablated Lb female wasps. Source data are provided as a Source Data file.

---

impaired host search performance was due to a reduced complement of eggs, we examined the ovary pairs in both young (4-day-old) and old (12-day-old) female parasitoid wasps. After dissection, we found that the ovary sizes and the mature egg numbers of young and old Lb females were comparable (Supplementary Fig. 3a, b). Collectively, the differences in search performance, such as the SI and locomotion speed, between the young and old Lb females might account for the decline in parasitic efficacy, which in turn affects decreased *Drosophila* oviposition.

We further monitored the search performance for *Orco* RNAi-treated young Lb female wasps and antenna-ablated young Lb female wasps since they present low parasitic efficiency (Fig. 2c). We also tested another kind of nonparasitic Lb wasp, ovipositor-ablated young female wasps. The SI values were significantly lower for the *Orco* RNAi-treated female wasps than the young females and were negligible for the antenna-ablated female wasps (Fig. 3a). Correspondingly, the trajectory analysis showed that the *Orco* RNAi-treated female wasps and antenna-ablated female wasps moved less than the young female wasps on fly food medium (Fig. 3b, c). In addition, the relevant locomotion speed of the *Orco* RNAi-treated female wasps and the antenna-ablated female wasps was also lower than that of the young Lb females (Fig. 3d). Strikingly, the ovipositor-ablated female wasps showed a normal SI value, locomotion trajectory and speed, which were comparable to those of the young female wasps (Fig. 3a–d). However, although the ovipositor-ablated Lb female wasps failed to parasitize the *Drosophila* host larvae, there was a similar defect in oviposition as that with the young female wasps (Fig. 3e, f). These results indicate that the search performance of female wasps is possibly responsible for the defensive response of *Drosophila*.

## Visualization of the parasitoid wasp is responsible for decreased oviposition

It has been reported that vision is necessary for flies to initiate the defensive response to accelerate sexual behavior when encountering parasitic wasps[25]. We next tested whether visual inputs were also responsible for the alternation of oviposition rates. We found that *GMR-grim* flies, which express an apoptotic activator in the developing retina leading to blindness[33], exhibited no oviposition changes when exposed to Lb females (Fig. 4a). In contrast, two independent *Orco* mutants (*Orco*[1] and *Orco*[2]), which fail to respond to most olfactory stimuli[34], initially had reduced oviposition rates in the presence of female wasps, but rates gradually returned to normal, as was observed with the wild-type flies (Fig. 4b, Supplementary Fig. 4a). To further elucidate the role of vision in the decreased oviposition response, we found that a visually impaired mutant, *ninaB*[1], showed equivalent egg laying in the presence and absence of Lb females (Supplementary Fig. 4b). We also performed the oviposition experiments in darkness (Fig. 4c). Oviposition rate decreases were not observed when the flies

cohabited in darkness with either young (4-day-old) or old (12-day-old) female parasitoid wasps (Fig. 4d), confirming the findings that vision is very important to the decreased egg laying. We next placed the flies and wasps in a special apparatus (see methods and Fig. 4e) to physically separate the two populations but allowed them to see each other through a transparent window. Oviposition was significantly suppressed in the wild-type flies that could see the young female wasps but not in the flies that could see the old wasps (Fig. 4f). We note that the effect of egg laying in this special apparatus (20% reduction) is smaller than that in regular fly bottles (44% reduction), leading us to propose the possible explanation that the visual cues from wasps in the transparent chamber were much weaker. However, it is also possible that some other sensory modalities (e.g., olfaction and audition) can strength the visual-induced egg-laying reduction, while they are not sufficient alone.

Lobular columnar (LC) neurons in the lobula are responsible for transmitting primary visual information to higher brain regions[35,36]. Silencing of one class of LC neurons, LC4, has previously been shown to reduce the mating acceleration response of *Drosophila* to parasitoids[25]. We then used LC4-specific split GAL4 lines driving the *UAS-TNT* transgene (tetanus toxin light chain, which cleaves synaptobrevin to block synaptic transmission) to block LC4 neuron activity. We found that parental control flies containing a *UAS-TNT* transgene alone or split-GAL4 constructs "*LC4-GAL4*" alone showed the expected reduction in egg laying when exposed to 4-day-old (young) Lb females (Fig. 4g). However, exposed flies in which LC4 neurons were blocked (*LC4-GAL4 > UAS-TNT*) did not show a reduction in egg laying. When thermally activated transient receptor potential channel A1 (*UAS-TRPA1*) was ectopically expressed under the control of *LC4-GAL4* to conditionally increase the activity of LC4 neurons, we found that the TrpA1 activation of LC4 neurons did not induce the egg reduction behavior (Supplementary Fig. 5). These results show that these neurons are necessary but not sufficient to initiate the effect of young female wasps (Fig. 4g; Supplementary Fig. 5).

Taken together, these results indicate that *D. melanogaster* females depend on LC4 visual projection neurons to sense parasitoid female wasps in their environment and the visual cues from Lb females (i.e., the search performance) initiate changes in oviposition.

## Exposure to wasps inhibits fly ovulation and induces egg retention

To identify the underlying mechanisms that change the oviposition rate in the presence of a threat from wasps, we examined the ovary pairs in both wasp-exposed and unexposed female flies. As with the previous approaches, the flies in the group exposed to young female wasps laid fewer eggs on Day 2 than those in the unexposed group, but the oviposition rate of the exposed flies was normal on Day 10 (Fig. 1b). After dissection, we found that the ovaries of the exposed flies were

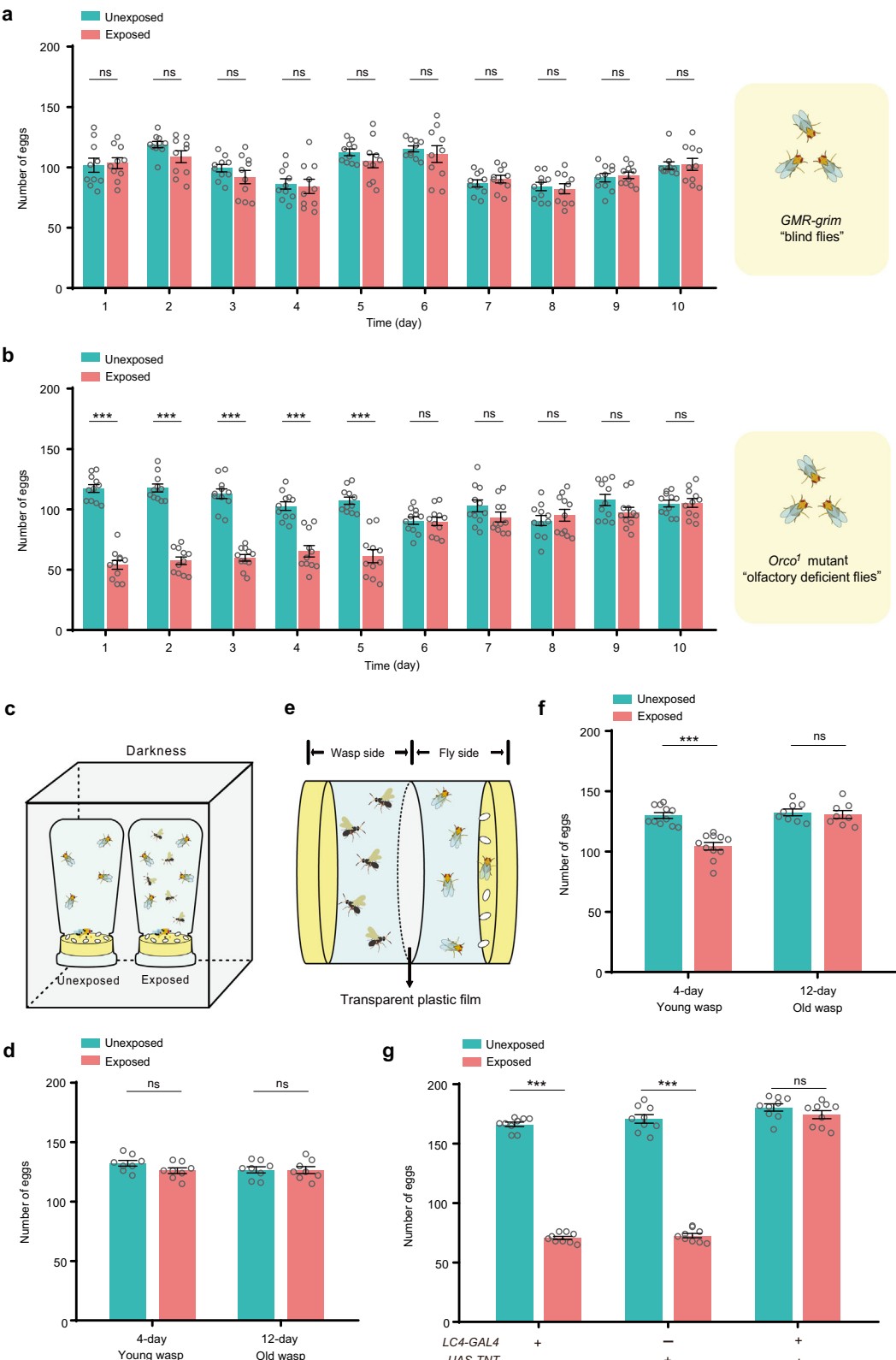

much larger than those of the unexposed controls on Day 2 (Supplementary Fig. 6a–c). As expected, the ovary sizes were comparable on Day 10, since both the unexposed and exposed female flies laid similar numbers of eggs (Supplementary Fig. 6a–c). Moreover, ovary enlargement in the exposed flies on Day 2 was caused by increased ovarian retention of mature follicles, as these ovaries contained more mature eggs per ovary than those of the unexposed flies (Fig. 5a, b). These results indicate that the reduction in egg laying is probably due to an arrest in ovulation.

During ovulation, mature eggs (stage 14 egg chambers, also known as mature follicles or mature oocytes) are released from the ovary into the oviduct and subsequently into the uterus. This process requires active proteolytic degradation of the follicle wall and follicle rupture[37]. Specifically, posterior follicle cells of a mature egg chamber

**Fig. 4 | Decreased oviposition depends on visual cues. a** The daily number of eggs of blind *GMR-grim* flies exposed to Lb female wasps compared to unexposed flies. Blind flies were exposed to wasps for a testing period of 10 days. The experiment was performed ten times. Data represent the mean ± SEM. Significance was determined by two-way ANOVA with Sidak's multiple comparisons test, *p* values are indicated in Source Data file (ns, not significant). **b** The daily number of eggs of olfactory-deficient *Orco[1]* mutant flies exposed to Lb female wasps compared to that of unexposed flies. Olfactory-deficient flies were exposed to wasps for a testing period of 10 days. The experiment was performed eleven times. Data represent the mean ± SEM. Significance was determined by two-way ANOVA with Sidak's multiple comparisons test, *p* values are indicated in Source Data file (***p* < 0.001; ns, not significant). **c** Oviposition experiment setup for the dark conditions. **d** Egg numbers of CS flies exposed to 4-day-old (young) or 12-day-old (old) Lb females compared with unexposed flies under dark conditions. Eggs were counted on Day 2. The experiment was performed eight times. Data represent the mean ± SEM.

Significance was determined by two-way ANOVA with Sidak's multiple comparisons test, *p* values are indicated in Source Data file (ns, not significant). **e** Experimental setup for visual exposure only of flies and wasps. **f** Egg numbers of CS flies only visually exposed to 4-day-old (young) or 12-day-old (old) Lb females compared with unexposed flies. Eggs were counted on Day 2. The experiment was performed at least eight times. Data represent the mean ± SEM. Significance was determined by two-way ANOVA with Sidak's multiple comparisons test, *p* values are indicated in Source Data file (***p* < 0.001; ns, not significant). **g** The number of eggs laid by the unexposed and exposed *D. melanogaster* females, including *UAS-TNT*, *LC4-GAL4*, and *LC4-GAL4 > UAS-TNT* genotypes. Eggs were counted on Day 2. The experiment was performed nine times. Data represent the mean ± SEM. Significance was determined by two-way ANOVA with Sidak's multiple comparisons test, *p* values are indicated in Source Data file (***p* < 0.001; ns, not significant). Source data are provided as a Source Data file.

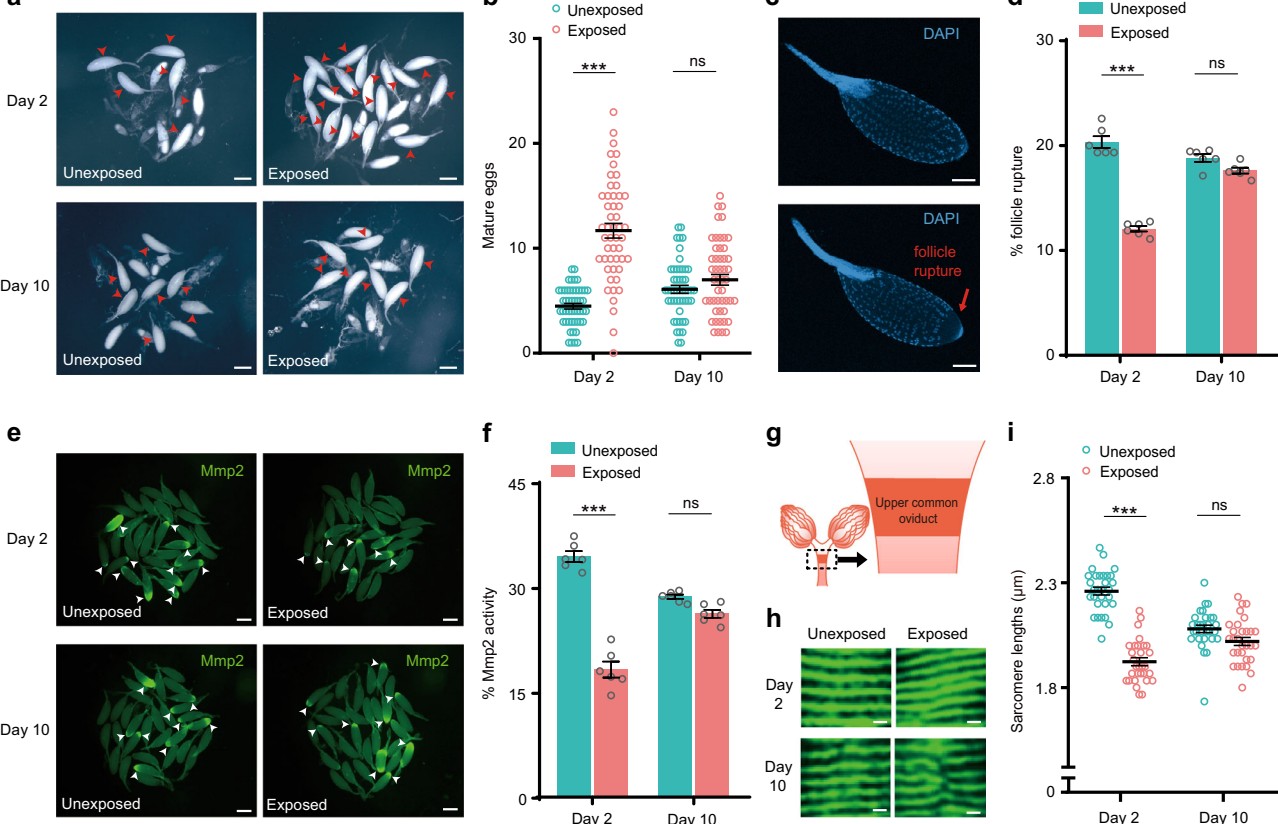

**Fig. 5 | Exposure to wasps decreases follicle rupture and Mmp2 activity. a** Images of mature eggs (red arrows) from single pairs of ovaries of the unexposed and exposed *Drosophila* on Day 2 and Day 10. Scale bars, 400 μm. **b** The number of mature eggs per ovary in the unexposed and exposed flies on Day 2 and Day 10. *n* = 50 per group. Data represent the mean ± SEM. Significance was determined by two-way ANOVA with Sidak's multiple comparisons test, *p* values are indicated in Source Data file (***p* < 0.001; ns, not significant). **c** Images of DAPI-stained mature eggs with partial follicle rupture (red arrow) and oocytes fully covered by follicular cells. Six biologically independent experiments were performed. Scale bars, 100 μm. **d** Percent of mature eggs with follicle rupture of the unexposed and exposed flies on Day 2 and Day 10. *n* = 6 per group; ~30 mature oocytes per replicate. Data represent the mean ± SEM. Significance was determined by two-way ANOVA with Sidak's multiple comparisons test, *p* values are indicated in Source Data file (***p* < 0.001; ns, not significant). **e** In situ zymography images of Mmp2 activity (green) in mature eggs of ovaries of the exposed and unexposed flies

collected on Day 2 and Day 10. Egg chambers with posterior Mmp2 activity are marked by white arrowheads. Scale bars, 400 μm. **f** Percent of eggs with Mmp2 activity of the unexposed and exposed flies on Day 2 and Day 10. *n* = 6 per group; ~30 mature oocytes per replicate. Data represent the mean ± SEM. Significance was determined by two-way ANOVA with Sidak's multiple comparisons test, *p* values are indicated in Source Data file (***p* < 0.001; ns, not significant). **g** Cartoon of the *Drosophila* ovary and oviduct. The focus region in the upper common oviduct (dashed box) is indicated. **h** Representative images of single myofibrils from oviducts of the unexposed and exposed groups. The distance between each GFP band represents the sarcomere length. Three biologically independent experiments were performed. Scale bars, 2 μm. **i** Sarcomere lengths in focus regions of oviducts of the unexposed and exposed females on Day 2 and Day 10. *n* = 30 per group. Data represent the mean ± SEM. Significance was determined by two-way ANOVA with Sidak's multiple comparisons test, *p* values are indicated in Source Data file (***p* < 0.001; ns, not significant). Source data are provided as a Source Data file.

are trimmed, breaking the follicle-cell layer and allowing the egg to be released into the oviduct[38,39]. We further investigated whether this process was impaired in the exposed flies using two different methods (see details in the Methods). First, samples were DAPI stained, and the ratio of mature oocytes fully or partially covered by follicular cells was quantified (Fig. 5c). As expected, posterior trimmed follicles were readily observed in the ovaries of the controls, accounting for 20% and 19% of the total mature follicles in the unexposed control flies on Day 2 and Day 10, respectively (Fig. 5d). The percent of posterior trimmed follicles was reduced significantly in the females exposed to young Lb female wasps for 2 days, but the percent was similar in the exposed and unexposed flies on Day 10 (Fig. 5d). Second, we used *R47A04-Gal4* to drive the expression of *UAS-RFP* specifically in follicle cells[37]. By monitoring the RFP signal, we also observed a decrease in follicle trimming of mature oocytes in the flies exposed to wasps for 2 days but not for 10 days compared to the unexposed controls (Supplementary Fig. 7a, b).

Matrix metalloproteinase 2 (Mmp2) is an enzyme that is required for trimming posterior follicle cells and plays a crucial role in facilitating ovulation[40]. We thus speculated that the activity of Mmp2 contributes to the egg-laying arrest in flies that occurs after exposure to wasps. We carried out in situ gelatinase assays to measure Mmp2 activity within follicles of the exposed and unexposed flies (Fig. 5e). Approximately 35% and 29% of the mature follicles had gelatinase activity at their posterior end in the unexposed flies on Day 2 and Day 10, respectively (Fig. 5f). In contrast, 18% of the mature follicles in the exposed flies had gelatinase activity on Day 2, a significant decrease compared to the unexposed controls (Fig. 5e, f). Mmp2 activity did not differ in the exposed flies and the unexposed flies on Day 10 (Fig. 5e, f). Together, our data indicate that Mmp2 activity is significantly decreased, which in turn impairs the rupture of follicular cells around mature oocytes with a subsequent reduction in the number of eggs laid when flies are exposed to young parasitic wasps.

## Exposure to wasps stimulates the contractions of *Drosophila* oviduct muscle

The *Drosophila* oviduct contains circular striated muscle fibers but no longitudinal muscle fibers[41]. The relaxation of oviduct muscles facilitates the movement of eggs from the ovary into the oviduct[42]. We speculated that exposure to wasps also affects the relaxation of the oviduct in flies. To test this hypothesis, we exposed female flies that express the myosin heavy chain fused to GFP (MHC-GFP)[43] to Lb females. In these flies, the distance between each GFP band reflects the sarcomere length. As such, we can easily examine oviduct muscle tonus by measuring the average sarcomere length at the upper common oviduct (Fig. 5g). Sarcomere lengths were significantly shorter in the exposed flies than in the unexposed controls on Day 2 (Fig. 5h, i). In contrast, sarcomere lengths were indistinguishable in the exposed and unexposed flies on Day 10 (Fig. 5h, i). We next investigated whether the eggs became stuck in the oviduct due to abnormal muscle contraction. To our surprise, there was no obvious difference in the oviduct with the stuck eggs between the exposed flies and the unexposed controls (Supplementary Fig. 8). These results indicate that the presence of young wasp females causes dysfunctional contractions of oviduct muscles, but this change may not be the key factor to suppress egg laying in exposed flies.

## Exposure to wasps decreases *Drosophila* egg laying through OA neuronal signaling

Approximately 70-100 octopaminergic neurons are dispersed throughout the *Drosophila* nervous system and produce the octopamine (OA), which is an important neuromodulator[44]. In the ventral nerve cord (VNC) region, there are five different octopaminergic neuron clusters based on their position, which include PTS (single cell

in the midline of the prothoracic neuromere), PTC (cell cluster in the midline of the prothoracic neuromere), MSC (cell cluster in the midline of the mesothoracic neuromere), MTC (cell cluster in the midline of the metathoracic neuromere) and AC (cell cluster in the thoracic abdominal ganglia)[45,46] (Fig. 6a). Importantly, AC neurons in the VNC region innervate female reproductive tissues such as the ovaries, oviducts, and uterus[47] and modulate OA-dependent egg-laying behaviors, including mature follicle trimming, rupture and oviduct muscle relaxation[39,41,48] (Fig. 6a). Therefore, our above results have suggested that OA is possibly involved in the decreased oviposition response in the presence of young Lb females (Fig. 5; Supplementary Fig. 7). OA is synthesized from tyrosine by the sequential actions of tyrosine decarboxylase 2 (Tdc2) and tyramine beta-hydroxylase (Tβh)[49,50]. We then analyzed the expression of the mRNAs encoding Tdc2 and Tβh in the brain and VNC of the exposed and unexposed *Drosophila* females by qRT-PCR. Strikingly, no significant difference was observed in the brains of the exposed flies compared to the unexposed female flies after exposure to young Lb wasps on Day 2 or on Day 10 (Fig. 6b, c). However, the levels of *Tdc2* and *Tβh* were significantly decreased in VNC when the female flies were exposed to wasps for 2 days but not for 10 days (Fig. 6d, e). We also examined the expression levels of *Tdc2* and *Tβh* in the *GMR-grim* flies. There was no difference between the exposed and unexposed *GMR-grim* flies in VNC after a 2-day exposure to young female wasps (Supplementary Fig. 9). These results indicate that the expression of two key genes (*Tdc2* and *Tβh*) is downregulated specifically in the VNC region of *Drosophila* females when they see active natural enemies.

Decreased levels of *Tdc2* and *Tβh* upon threat exposure are expected to reduce both OA and tyramine levels, and OA is widely reported to regulate insect egg-laying behaviors[37,47,48,50,51]. We then used whole-mount immunohistochemistry to examine the locations and levels of OA in the VNC of the exposed and unexposed *Drosophila* females. Antibody staining supports that OA in VNC is produced specifically in different octopaminergic neuron clusters, including PTS, PTC, MSC, MTC and AC (Fig. 6f). Consistent with the qRT-PCR results, we found that the levels of OA were significantly decreased in all octopaminergic neurons of the VNC after a 2-day exposure to young female wasps (Fig. 6f, g). We next examined whether OA levels were impaired in the axons of octopaminergic neurons on the reproductive tracts of exposed *Drosophila* females. Immunohistochemistry analyses showed that OA-immunoreactive nerve termini were found in all regions of the reproductive tract, including the ovary (OV), lateral oviducts (LO), upper common oviduct (COU), and lower common oviduct (COD) (Fig. 7a–c). The highest fluorescence intensity for OA was in nerve termini of the LO, followed by COU and COD (Fig. 7c). The results further showed that OA intensity was significantly decreased in all different regions in the reproductive tracts of exposed flies compared to unexposed female flies after exposure to young Lb wasps on Day 2 (Fig. 7b, c).

To further investigate whether ectopically increasing the activity of OA neurons could rescue the egg-laying defect of the exposed *Drosophila* females, we carried out two independent experiments with the help of *Tdc2-GAL4*. This *GAL4* line is specifically expressed in OA- and tyramine-producing neurons and has been widely used to manipulate OA neuronal activity in vivo[49,52,53]. First, we ectopically expressed *UAS-TRPA1* under the control of *Tdc2-GAL4* to conditionally increase the activity of OA neurons[54,55] (Fig. 7d). At the nonactivating temperature of 23 °C, egg-laying defects were observed in both the *Tdc2-GAL4* and *UAS-TRPA1* control fly females and in the *Tdc2-GAL4 > UAS-TRPA1* females after 2 days of exposure to young Lb females (Fig. 7d). However, at the TRPA1-activating temperature of 29 °C, the wasp-exposed *Tdc2-GAL4 > UAS-TRPA1* females, which have high levels of OA neuron activity, laid significantly more eggs than the wasp-exposed control females (Fig. 7d). Next, we used the *Tdc2-GAL4* driver to express a gene (*UAS-eag^{DN}*) encoding a dominant-negative

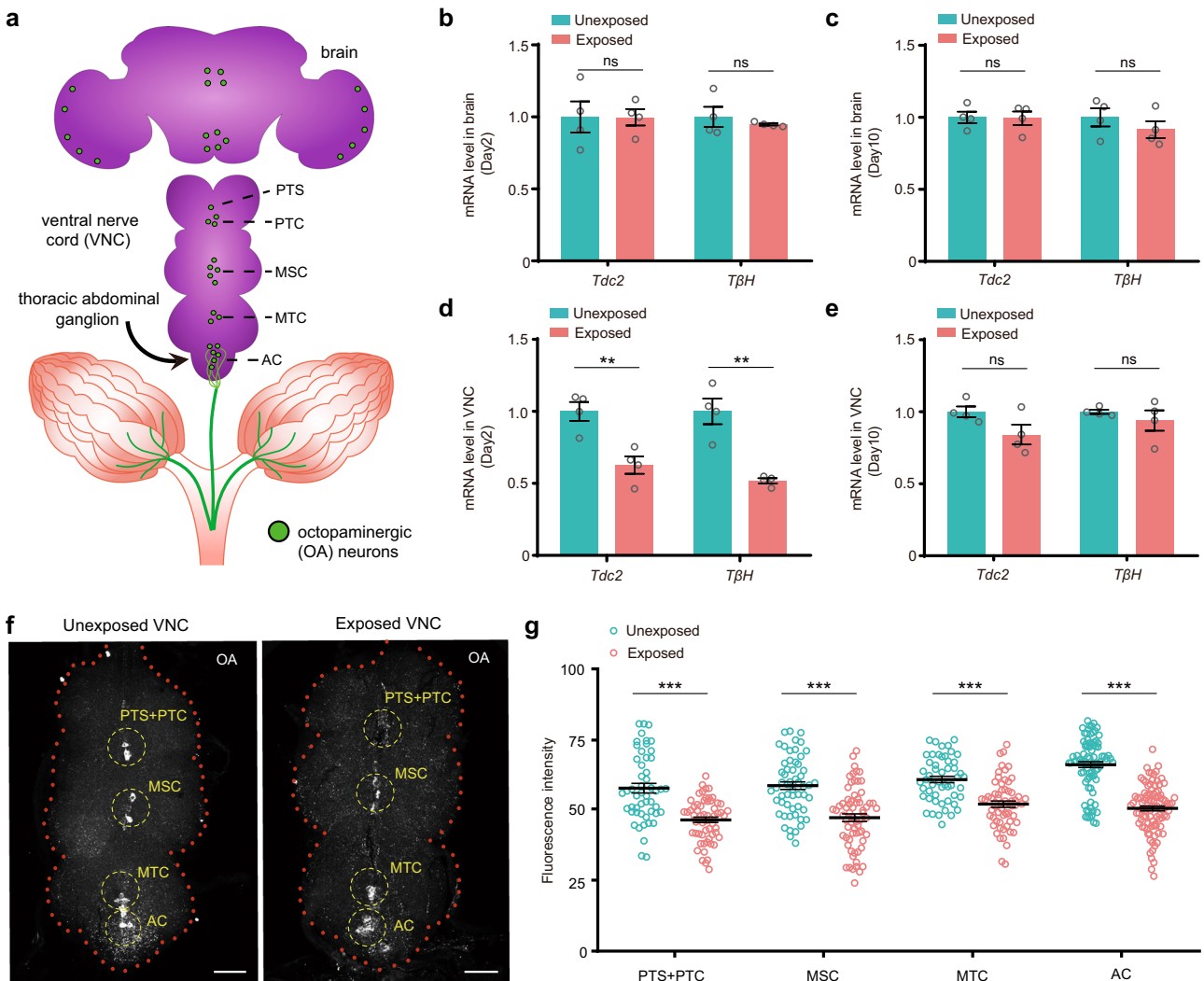

**Fig. 6 | Exposure to wasps decreases OA production in VNC. a** Cartoon of the female reproductive tract and the brain, VNC, and thoracic abdominal ganglion. Green spots represent the OA neurons. PTS, single cell in midline of the prothoracic neuromere; PTC, cell cluster in midline of the prothoracic neuromere; MSC, cell cluster in midline of the mesothoracic neuromere; MTC, cell cluster in midline of the metathoracic neuromere; AC, cell cluster in the thoracic abdominal ganglia. **b** Quantification of *Tdc2* and *Tβh* mRNA levels in the brains of the unexposed and exposed female flies on Day 2. *n* = 4 per group. Data represent the mean ± SEM. Significance was determined by two-sided unpaired Student's *t* test, *p* values are indicated in Source Data file (ns, not significant). **c** Quantification of *Tdc2* and *Tβh* mRNA levels in the brains of the unexposed and exposed female flies on Day 10. *n* = 4 per group. Data represent the mean ± SEM. Significance was determined by two-sided unpaired Student's *t* test, *p* values are indicated in Source Data file (ns, not significant). **d** Quantification of *Tdc2* and *Tβh* mRNA levels in the VNC of the unexposed and exposed female flies on Day 2. *n* = 4 per group. Data represent the

mean ± SEM. Significance was determined by two-sided unpaired Student's *t* test, *p* values are indicated in Source Data file (**$p < 0.01$). **e** Quantification of *Tdc2* and *Tβh* mRNA levels in the VNC of the unexposed and exposed female flies on Day 10. *n* = 4 per group. Data represent the mean ± SEM. Significance was determined by two-sided unpaired Student's *t* test, *p* values are indicated in Source Data file (ns, not significant). **f** Representative images of OA (white) immunolocalization in the unexposed and exposed VNC of female flies on Day 2. Three biologically independent experiments were performed. Scale bars, 50 μm. **g** Fluorescence intensity of OA immunolocalization in the unexposed and exposed octopaminergic neurons of the indicated clusters of female flies on Day 2. Plotted is the mean intensity from 3 areas within each cell. Left to right, *n* = 51, 60, 56, 66, 52, 62, 84, and 96. Data represent the mean ± SEM. Significance was determined by two-way ANOVA with Sidak's multiple comparisons test, *p* values are indicated in Source Data file (***$p < 0.001$). Source data are provided as a Source Data file.

ether-a-gogo potassium channel subunit in OA neurons[56]. Loss of function of eag results in increased neuronal activity[48]. As expected, the control *Tdc2-GAL4* and *UAS-eag^DN* females laid significantly fewer eggs after exposure to wasps for 2 days than the unexposed flies (Fig. 7e). However, the *Tdc2-GAL4 > UAS-eag^DN* females, which have elevated activity of OA neurons, showed a significant increase in oviposition rate relative to the controls that were exposed to young Lb females (Fig. 7e). These results indicate that increasing OA neuronal activity in *Drosophila* females compensates for the oviposition deficiency induced by the presence of young female wasps.

We then examined whether injection of OA would stimulate egg laying in flies exposed to wasps. As expected, the OA-injected individuals had an increased oviposition rate compared to the controls injected with doubly distilled $H_2O$ when they were exposed to young Lb female wasps (Fig. 7f). These results further support our conclusion that the reduced oviposition rate of wasp-exposed *Drosophila* females can be rescued by elevating OA neuronal activity.

Taken together, the results indicate that a reduction in OA neuronal signaling in *D. melanogaster* females triggered by visual cues in the presence of young Lb wasps results in depression of oviposition.

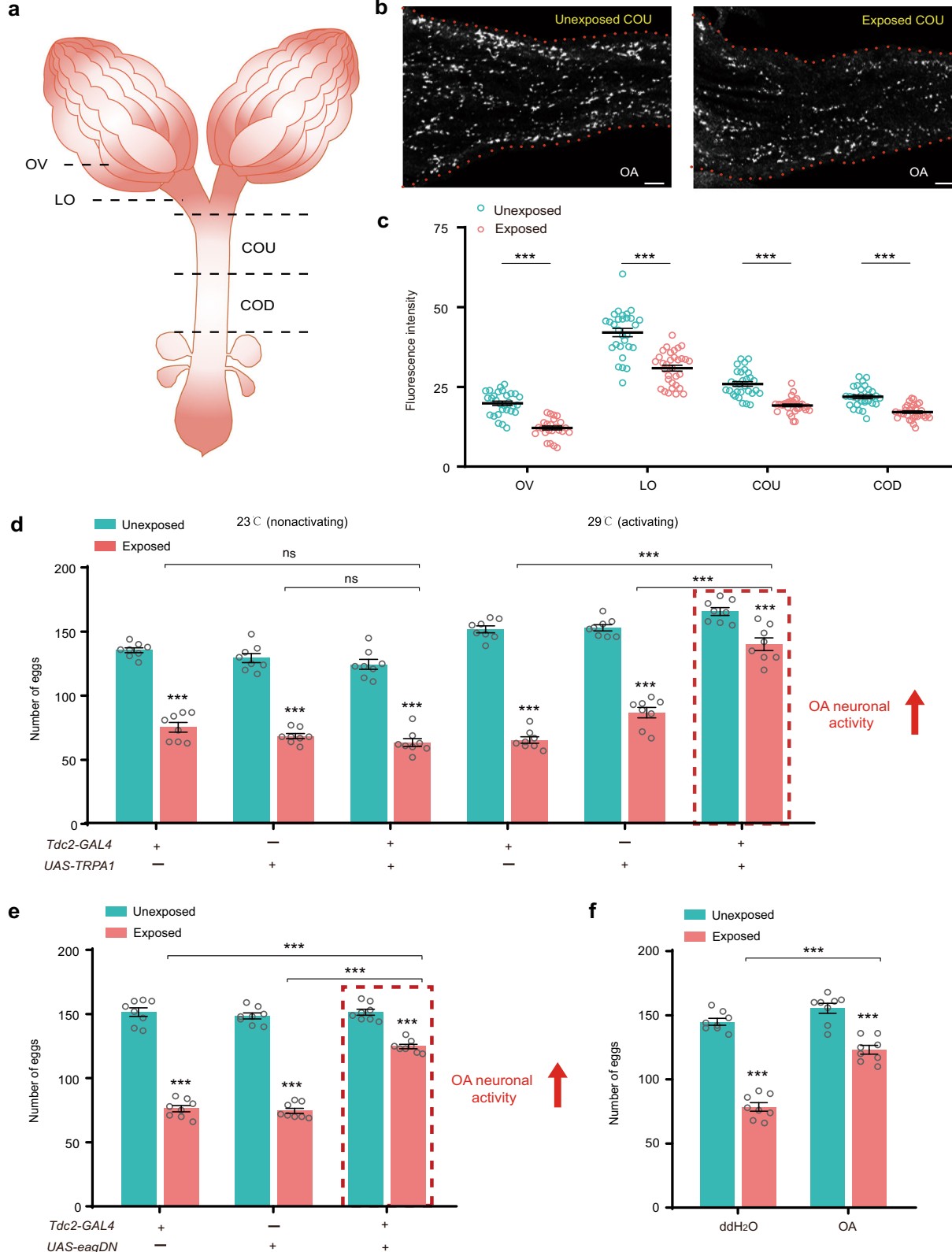

## Two neuronal signaling pathways underlie the diverse defensive behavior responses to the same parasitoid threats

In addition to depression of oviposition, *Drosophila* females also switch to laying their eggs on ethanol-laden food when they encounter deadly parasitic wasps[23]. This strategy is thought to be a behavioral immune response that protects hatched offspring from infection. The ethanol preference in *D. melanogaster* is linked to a decrease in neuropeptide F (NPF) in the brain in the presence of parasitic wasps[23]. We therefore sought to determine whether NPF also plays a role in depressed oviposition responses and whether NPF and OA neuronal signaling have direct causal connections. We inhibited the expression of NPF using a shRNA expressed from the *NPF-GAL4* driver, which is specifically expressed in all NPF-producing neurons. As expected, a significant decrease in NPF levels was found when fly brains were

**Fig. 7 | OA mediates parasitic wasp-induced suppression of oviposition.**
**a** Cartoon of the female reproductive tract regions, including the ovary (OV), lateral oviduct (LO), upper common oviduct (COU), and lower common oviduct (COD). **b** Representative image of OA (white) immunolocalization in the unexposed and exposed COU regions of female flies on Day 2. Three biologically independent experiments were performed. Scale bars, 10 µm. **c** Fluorescence intensity of OA immunolocalization in the unexposed and exposed female reproductive tract regions of female flies on Day 2. Plotted is the overall mean intensity for each specific region. Left to right, $n = 29, 25, 29, 32, 30, 32, 30$, and $32$. Data represent the mean ± SEM. Significance was determined by two-way ANOVA with Sidak's multiple comparisons test, $p$ values are indicated in Source Data file (***$p < 0.001$). **d** The number of eggs laid by the unexposed and exposed *D. melanogaster* female flies, including *Tdc2-GAL4*, *UAS-TRPA1*, and *Tdc2-GAL4 > UAS-TRPA1* flies of different genotypes. Eggs were counted on Day 2. The temperature of 23 °C is nonactivating, and 29 °C increases the activity of OA neurons. The experiment was performed

eight times. Data represent the mean ± SEM. Significance was determined by two-way ANOVA with Sidak's multiple comparisons test, $p$ values are indicated in Source Data file (***$p < 0.001$; ns, not significant). **e** The number of eggs laid by unexposed and exposed *D. melanogaster* female flies, including *Tdc2-GAL4*, *UAS-eag^DN*, and *Tdc2-GAL4 > UAS-eag^DN* flies of different genotypes. Eggs were counted on Day 2. Overexpression of eag^DN increases OA neuronal activity. The experiment was performed eight times. Data represent the mean ± SEM. Significance was determined by two-way ANOVA with Sidak's multiple comparisons test, $p$ values are indicated in Source Data file (***$p < 0.001$). **f** The number of eggs laid by unexposed and exposed flies 1 day post OA or doubly distilled $H_2O$ (dd$H_2O$) injection. The experiment was performed eight times. Data represent the mean ± SEM. Significance was determined by two-way ANOVA with Sidak's multiple comparisons test, $p$ values are indicated in Source Data file (***$p < 0.001$). Source data are provided as a Source Data file.

immunostained with NPF antiserum, indicating that *NPF* was successfully targeted for degradation (Supplementary Fig. 10a, b). The proportion of eggs laid by the *NPF-GAL4 > UAS-NPF RNAi* females was marginally reduced compared to that of the *NPF-GAL4* and *UAS-NPF RNAi* control females on several testing days (Supplementary Fig. 10c). We next evaluated follicle rupture and Mmp2 activity in flies with decreased NPF levels and observed no significant differences between the *NPF-Gal4 > UAS-NPF RNAi* and control females (Supplementary Fig. 10d, e). Similarly, when NPF receptor (*NPFR*) was specifically knocked down in OA neurons of *Drosophila* females, the oviposition rate, the percent of posterior trimmed follicles, and Mmp2 activity were not affected (Supplementary Fig. 11a–c). These data indicate that decreased NPF does not alter OA-mediated neuronal signaling to prevent *Drosophila* ovulation. Correspondingly, to address whether OA-mediated neuronal signaling influences *NPF* levels, we inhibited the expression of the four OA receptors in NPF-producing neurons[57–59]. There were no changes in *NPF* levels in OA signaling-ablated NPF neurons (Supplementary Fig. 12a–d).

Taken together, these observations indicate that the two neuronal signaling pathways act independently to elicit two different behavioral outputs in the presence of female wasps. A reduction in NPF signaling is mainly responsible for the ethanol preference response[23], whereas a reduction in OA signaling leads to decreased oviposition.

## Discussion

Animal survival depends on the ability to sense predators and generate behavioral and physiological defense responses[2–4,6–8]. Defense responses are assumed to incur large costs; thus, many organisms have evolved an adaptive strategy to accrue defenses by precisely classifying predatory threats according to the magnitude of risk associated with an encounter[10–12]. For instance, elephants can make subtle distinctions between language and voice characteristics to correctly identify the most threatening individuals[60]; vervet monkeys are well known to have the ability of predator classification and give different alarm calls for different predators[61]. However, few studies have investigated the mechanisms underlying these adaptive responses. *Drosophila* and its parasitoids provide an excellent model with which to study a broad variety of questions in ecology and evolution[25,62,63]. In this study, we discovered that *D. melanogaster* females laid fewer eggs in the presence of parasitoid wasps. We further found that decreased oviposition was maintained for approximately 6 days in the presence of Lb female wasps, and then, the oviposition rates reverted to the unexposed levels. Lb is a short-lived parasitoid[64], and the parasitic ability significantly decreases with wasp age. Old wasps represent little threat to *Drosophila* larvae. Therefore, this strategy of switching from decreased oviposition to normal egg laying increases the survival of offspring when it is difficult or costly for *D. melanogaster* females to leave the wasp-infected environment and find a more favorable egg-laying environment.

By manipulation of the environment and by use of Lb and *Drosophila* mutants and/or transgenics, we further demonstrated that visual cues via visual projection neurons (VPNs) allow female flies to determine how dangerous parasitic wasps are to their offspring larvae. VPNs transmit information from the optic lobe to higher brain regions. The most numerous VPNs in the *Drosophila* lobula are LC neurons, which contain multiple types on their anatomy[35,36]. Previous studies have shown that at least one type of *Drosophila* LC neurons, LC4, is activated by the looming stimulus of their predators. In turn, these LC4 neuron cells synapse onto the giant fiber descending neurons to induce a quick jump escape response[36,65,66]. It is striking that the LC4 neurons contribute to both wasp-induced oviposition reduction and looming stimulus-induced escape response. Our results show that the reduction of egg laying in the transparent chamber is weaker than that in regular fly bottles, potentially reflecting the different effects derived from far and close visual stimulus (Fig. 4f). Based on these results, it seems possible that the parasitic wasps may present some certain of looming stimulus to flies. In addition, the functional relevance of the different visual cues in LC4 cells and their downstream signals remains to be determined.

Our data indicate that *D. melanogaster* females observed host search performance to detect dangerous wasps, leading to the defensive oviposition response. *Drosophila* females did not decrease egg laying in the presence of old female wasps, young *Orco RNAi*-treated wasps, or antenna-ablated female wasps, which all have decreased host search performance, i.e., a low search index and decreased locomotion distance and speed, resulting in low or no parasitic ability. Ovipositor-ablated young female wasps do not have the ability to parasitize *Drosophila* larvae; however, search performance was normal, and the *Drosophila* females showed a reduction in oviposition rate when they cohabited with the ovipositor-ablated female wasps. The ability to derive information about a predator's behavior has been reported in some other prey species[67], and this could be a widespread adaptation. However, it is still not known whether the host search performance of female wasps is also responsible for other well-known antiparasitoid responses, e.g., ethanol-laden food preference[23] and accelerated sexual behavior[25]. Accordingly, it will be of interest to determine whether flies can elevate predation risk levels to undertake appropriate defensive responses since previous studies have been performed only with young female wasps[23,25].

We also identified the neuronal signals underlying the manipulation of the *Drosophila* oviposition response to Lb female wasps. Each *D. melanogaster* female has a pair of ovaries, and each ovary contains approximately 15-18 ovarioles. The egg chambers are assembled in the germarium at the anterior of the ovariole and develop through 14 different stages toward the posterior end[68]. Mature eggs are released from the ovary into the oviduct and subsequently into the uterus and oviposited onto the food substrate. The hormonal and neuronal controls of egg-laying behavior are well characterized[39,42,48,69,70]. Most

importantly, the biogenic amine OA is critical for the ovulation process in *Drosophila* egg laying. OA not only regulates ovulation by inducing the relaxation of oviduct muscle but also directly stimulates Mmp2 activity to trigger the rupture of mature follicles, which releases fertilizable eggs into the oviduct[37,40,71]. Dysregulation of OA-mediated signaling causes severe ovulation disorders. In this study, we found that OA neuronal signaling activity was decreased dramatically when *Drosophila* females cohabited with young Lb females but not old wasps. Specifically, we observed reduced expression of *Tdc2* and *Tβh* in the OA biosynthesis pathway, decreased Mmp2 activity, subsequent impaired trimming of mature follicles, dysfunctional contraction of oviduct musculature, and retention of mature eggs in females that cohabited with young wasps. Though it is still not clear how the visual inputs reduce the expression of *Tdc2* and *Tβh* limited in VNC region, our data indicate that OA neuronal signaling is involved in the decreased oviposition response of *Drosophila* to their parasitoids (Fig. 8).

Although we focused on the principal ovulation regulator, OA, it is likely that other signaling factors are involved in this response to predation because the elevation of OA neuronal activity or OA quantity partially (but not fully) increases oviposition rate of exposed *Drosophila* females. A decreased level of NPF in the *Drosophila* female brain is responsible for ethanol-laden food preference[23]. Strikingly, we found that *Drosophila* females marginally reduced the oviposition rate when the NPF levels were reduced. Thus, it is possible that NPF signaling has some contribution to the reduced oviposition in the presence of wasp females. However, our results support that NPF signaling functions independently from OA-mediated neuronal signaling. Because the decreased oviposition and ethanol food preference elicited by the presence of female wasps all require an intact *Drosophila* visual system, it will be interesting and urgently necessary to identify the specific neural circuits that begin in the retina and separately mediate the OA and NPF signaling pathways in future studies.

In summary, our study reveals the molecular mechanisms by which *Drosophila* females manipulate their oviposition rate via OA-mediated neuronal signaling under the stress of deadly parasitoid wasps. Our findings further indicate that *Drosophila* females have evolved to evaluate the degree of life-threatening stimuli by visually observing the host search performance of wasps (Fig. 8). These findings provide innovative insights into the mechanisms of defensive behaviors that are likely to be exploited by other prey animals in response to predators and are suggestive of a possibly conserved predation risk evaluation strategy.

## Methods

### Insects
The *D. melanogaster* strain *Canton-S* was used as a wild-type strain. *Canton-S* (BL64349), *Orco*[1] (BL23129), *Orco*[2] (BL23130), *GMR-grim* (BL52016), *ninaB*[1] (also known as *ninaB*[P315], BL24776), *R47A04-GAL4* (BL50286), *UAS-RFP* (BL30556), *MHC-GFP* (BL38462), *NPF-GAL4* (BL25682), *Tdc2-GAL4* (BL9313), and *UAS-eag*[DN] (BL8187) lines were acquired from the Bloomington Drosophila Stock Center. *UAS-NPF RNAi* (TH2569), *UAS-NPFR RNAi* (TH2116), *UAS-Oamb RNAi* (TH2000), *UAS-Octβ2R RNAi* (TH3666), and *UAS-Oct-Tyr RNAi* (TH2969) lines were acquired from Tsing Hua Fly Center. The *UAS-Octβ3R RNAi* line (31348R-4) was acquired from the NIG-FLY Center. The *UAS-TRPA1* and *UAS-TNT* lines were provided by Dr. Zefeng Gong (Zhejiang University, China), and the LC4-SS00315 line was provided by Dr. Yi Sun (Westlake University, China). *D. suzukii* was provided by Dr. Jia Huang (Zhejiang University, China). All flies were maintained on cornmeal/yeast/sugar *Drosophila* medium (the recipe can be found on Bloomington Drosophila Stock Center, https://bdsc.indiana.edu/information/recipes/bloomfood.html) at 25 °C, 50% humidity, and a 16 h:8 h light:dark cycle.

The *Drosophila* parasitoid wasps used in this study were *L. boulardi* (Lb), *L. heterotoma* (Lh), and *A. japonica* (Aj). Lb (G486) was

kindly provided by Dan Hultmark (Umeå University, Umeå, Sweden), Lh (Lh14) was kindly provided by István Andó (Biological Research Centre, Szeged, Hungary), Aj was collected from Taizhou (28°50′N, 120°34′E), Zhejiang, China in June 2018. All wasps were maintained on *D. melanogaster Canton-S* in our lab under the conditions as described[62,63]. Briefly, we placed 50 mated *Drosophila* females into a fly bottle containing cornmeal food. The flies were allowed to lay eggs for 2 h. After the *Drosophila* eggs hatched to the 2nd instar, approximately 48 h later, 10 mated parasitoid females (Lb, Lh, and Aj) were added to each bottle and allowed to parasitize the hosts for 6 h. The infected hosts were maintained at 25 °C until adult wasps emerged. The two nonparasitic wasps were *C. cunea* and *S. guani*, which were purchased from Jiyuan Baiyun, Inc., (Heinan, China) as pupae and used after eclosion. The newly emerged male and female wasps were collected and allowed to mate in vials with apple juice agar medium (27 g agar, 33 g brown sugar, and 330 ml pure apple juice in 1000 ml diluted water) for further use. All the wasps were not given any hosts to parasitize before starting the experiments and were never reused between experiments.

### *Drosophila* oviposition assay
Fly oviposition assay experiments were conducted using 177 ml plastic fly bottles (top: 3.5 cm diameter; bottom: 5.7 cm length and 5.7 cm width; height: 10.3 cm) (Fisher Scientific, Cat#11-888), which covered a dish (3.5 cm diameter) containing fly food medium (Fig. 1a). Newly emerged male and female *D. melanogaster* adults were collected and allowed to mate, and flies aged 3 days post eclosion were used for all experiments. Briefly, twenty 3-day-old female and five 3-day-old male *D. melanogaster* adults were placed into a fly bottle that either contained no wasps (unexposed) or contained wasps (exposed). The food dishes were replaced daily, and the eggs on the dishes were counted each day. Although dead flies or wasps seldom occurred during the oviposition assay experiments, the same number of flies or wasps at the same age were added as a substitute to the bottles from the rescue population that were prepared at the beginning of each experiment.

To determine whether decreased oviposition depended on visual inputs, we performed two additional assays. For assay where the acute response occurred in the dark, flies were anesthetized by ice and placed into fly bottles with or without Lb female wasps. The bottles were placed in a dark room, and flies were allowed to awaken in the dark. For the assay where the acute response occurred through vision without direct contact, a special apparatus (a cylindrical tube of 3.5 cm diameter and 2 cm height covered by fly food on both sides and with a transparent plastic film placed in the middle of the tube) was designed to separate the flies and female wasps into different chambers, and they were allowed to see each other through the transparent film (Fig. 4e).

Mated female wasps were used in all experiments, and wasps at different ages were used in the oviposition assay, including 2-day-old Lb females for long-term exposure experiments (Figs. 1; 4a, b; and Supplementary Figs. 2 and 4) and 4-day-old (young) and/or 12-day-old (old) Lb females for short-term exposure experiments (Figs. 2d; 3f; 4d, f, g; and 7d–f).

### Parasitic efficiency assay
Two-day-old mated Lb females were allowed to parasitize 2nd instar *Drosophila* larvae at parasite to host ratios of 1:20 for 3 h. The parasitized hosts were maintained at 25 °C until adult wasps emerged. The parasitism rate was calculated using the following formula:

$$\text{Parasitism rate}\,(\%) = (1 - \text{number of emerged \textit{Drosophila} adults} \,/\,\text{number of total hosts}) \times 100.$$

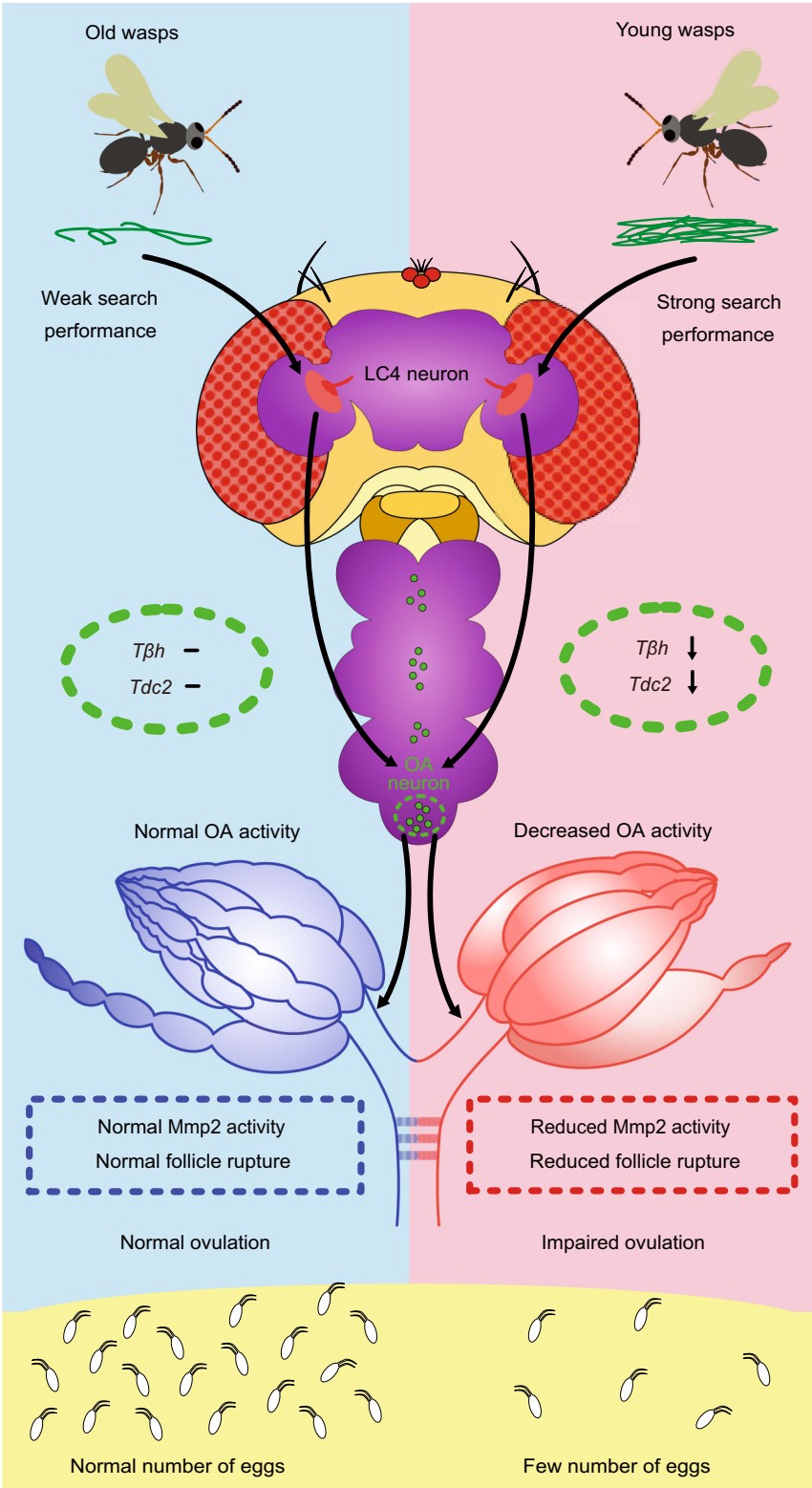

**Fig. 8 | The proposed model of decreased oviposition in response to young female wasps.** Young female wasps have a stronger search performance than old female wasps. When flies see their young parasitic wasps, they present an anti-parasitoid behavior to lay fewer eggs in response to visual perception through LC4 visual projection neurons. This oviposition depression is induced by the downregulation of the expression of *Tdc2* and *Tβh* in the ventral nerve cord, which in turn leads to the dramatic reduction of octopamine (OA). Then, the dysfunction of mature follicle trimming and rupture mediated by low levels of OA neuronal activity results in ovulation defects and a sharp decline in the oviposition rate.

### Host search performance analysis

The stereotyped search behavior of Lb females includes rhythmically drumming with their antennal tips, continuous movement on food substrate and frequent stinging with their sharp ovipositor in fly food (Supplementary Movie 1). We note that the *Leptopilina* parasitoids normally show host search behavior, no matter whether the fly larvae exist or not. The search index (SI), which is defined as the percent of time a wasp presents search behavior in a certain period of observation time (10 min in this study), was used as a measure of host search performance. Since the host search index can be very different if observed immediately after adding the wasps or a few hours later, we strictly monitored it within the same period after the release of a single wasp. The following types of wasps were assessed: 4-day-old Lb female wasps, 12-day-old Lb female wasps, 4-day-old *Orco RNAi*-treated Lb female wasps, 4-day-old antenna-ablated Lb female wasps, and 4-day-old ovipositor-ablated Lb female wasps. Specifically, after wasps were transferred into fly food bottles, they were allowed 30 min for acclimation to the new surroundings. After that, they were recorded by an Olympus Dp47 camera. The recording time for monitoring the SI was 10 min for each individual wasp in this study. It was scored using the following formula: SI (%) = time spent searching/total observing time x 100.

Locomotion trajectories (8 wasps were tracked at the same time) were determined for 4-day-old Lb female wasps, 12-day-old Lb female wasps, 4-day-old *Orco RNAi*-treated Lb female wasps, 4-day-old antenna-ablated Lb female wasps, and 4-day-old ovipositor-ablated Lb female wasps. Briefly, the wasps were transferred into a cylindrical tube of 2 cm diameter and 7.5 cm height covered by regular fly food. An Olympus Dp47 camera mounted directly above the arena was connected to a computer to record the wasp's search track for 2 min at a rate of 50 frames per second, and the position coordinates of the wasp in each frame and the relative locomotion speed were calculated using cellSens Dimension software (v2.2; Olympus). The locomotion distance was measured using ImageJ with Ridge (Line) Detection Plugin (v1.8.0; National Institutes of Health).

### Double-stranded RNA preparation and microinjection

Double-stranded RNA (dsRNA) was synthesized using the T7 RiboMAX Express RNAi System Kit (Promega, Cat# P1700) according to the manufacturer's instructions. The primers used, which are complementary to the Lb *Orco* gene, are listed in Supplementary Table 1. The reaction mixture was incubated at 37 °C for 4 h and heated at 70 °C for 10 min, and double strands were allowed to anneal at room temperature for 20 min. Subsequently, dsRNA was treated with RNase and DNase I to remove the templates and purified with isopropanol. The purified dsRNA was quantified with a NanoDrop 2000 (Thermo Fisher Scientific). Approximately 20 nl of dsRNA (5 μg/μL) was injected into fifth instar Lb larvae using the Eppendorf FemtoJet 4i device with the following parameters: injection pressure = 900 hPa; injection time = 0.15 s.

### Immunohistochemistry

*Drosophila* ovaries and VNCs were dissected in 1X PBS and fixed in 4% paraformaldehyde in PBS for 20 min, rinsed three times with 1X PBST (PBS containing 0.1% Triton X-100 and 0.05% Tween 20), and then blocked with 1% bovine serum albumin in 1X PBST for 1 h. Primary antibodies, including rabbit anti-OA (1:500; abcam, Cat# ab37092) and rabbit anti-NPF (1:500; gift from Dr. Zhangwu Zhao, China Agricultural University), were added and incubated overnight at 4 °C. After three 1X PBST washes, the tissues were incubated with Alexa 594 goat anti-rabbit secondary antibody (1:1000; Invitrogen, Cat# A-11012) for 2 h at room temperature. Samples were mounted in ProLong Gold Antifade Mountant with DAPI (Invitrogen, Cat# P36941). Fluorescence images were captured on a Zeiss LSM 800 confocal microscope.

ImageJ (v1.8.0; National Institutes of Health) was used to examine the fluorescence intensity level of immunoreactivity for OA. In the VNC region, the local mean fluorescence intensity level was determined, which quantifies the fluorescence intensity level within four different octopaminergic neuron clusters: PTS + PTC (single cell in the midline of the prothoracic neuromere and the cell cluster in the midline of the prothoracic neuromere), MSC (cell cluster in the midline of the mesothoracic neuromere), MTC (cell cluster in the midline of the metathoracic neuromere) and AC (cell cluster in the thoracic abdominal ganglia). Specifically, the intensity of 3 areas from a single neuron cell for each cluster was obtained to produce the mean intensity ($n \geq 50$ for each group). In reproductive tracts, the approach of overall mean fluorescence intensity level was conducted, which quantifies the average gray value within the different regions of reproductive tracts, e.g., ovary (OV), lateral oviduct (LO), upper common oviduct (COU) and lower common oviduct (COD).

### Ovary size and mature egg measurement

*Drosophila* females collected after Day 2 and Day 10 of the oviposition assay were dissected. The ovary images were taken under a stereoscope (Olympus MVX10) with a digital microscope camera (Olympus Dp47). The length and width of the ovaries from 60 unexposed and 64 exposed *Drosophila* females on Day 2 and the length and width of the ovaries from 63 unexposed and 63 exposed *Drosophila* females on Day 10 were measured by using ImageJ (v1.8.0; National Institutes of Health), respectively. The ovaries were dissected, and the numbers of mature eggs were counted for each individual *Drosophila* female. At least 50 female flies for each experimental group were dissected for mature egg measurement.

### Mature follicle trimming analysis

Follicle rupture analysis was carried out as described in published studies[37,40] with minor modifications. Briefly, *Drosophila* females from exposed and unexposed groups on Day 2 and Day 10 were frozen at −80 °C for 5 min, and ovaries were removed. The ovaries were immediately fixed with 4% paraformaldehyde and stained with DAPI. A trimmed follicle was scored according to the criteria that a quarter of the egg chamber at the posterior end has no follicle cell cover, and the number of mature follicles was scored according to their fully elongated dorsal appendage. The percent of trimmed follicles was then calculated by dividing the number of trimmed follicles by the number of mature follicles and multiplying by 100. The same procedure was performed to calculate follicle rupture based on RFP fluorescent signals driven by *R47A04-Gal4* and *UAS-GFP*. To avoid bias, the investigator was blinded to the origin of the mature eggs while counting the number of trimmed follicles. Six replicates were performed, and at least 30 mature follicles (randomly collected from 10 dissected *Drosophila* females) were evaluated for each replicate.

### Mmp2 activity measurement

The in situ zymography technique for gelatinase activity (Mmp2 activity) was performed as previously reported with minor modifications[40]. Ovaries were dissected in prewarmed Grace's media and incubated immediately in 100 μg/ml DQ-gelatin conjugated with fluorescein (Invitrogen, Cat# D12054) for 1 h. Ovaries were then fixed in 4% paraformaldehyde for 10 min. After three rinses in 1X PBS, mature follicles with posterior fluorescent signal were counted. To prevent observation bias, the investigator was blinded to the origin of the mature eggs while counting the fluorescent signal. The percent of follicles with Mmp2 activity was calculated by dividing the number of fluorescently labeled follicles by the number of total mature follicles and multiplying by 100. Six replicates were used, and at least 30 mature follicles (randomly collected from 10 dissected *Drosophila* females) were analyzed for each replicate.

## Oviduct muscle sarcomere length measurements

The sarcomere length in the upper common oviduct was measured as previously reported with minor modifications[48]. The reproductive tracts of wasp-exposed and unexposed *MHC-GFP Drosophila* females were dissected and immediately fixed in 4% paraformaldehyde for 20 min at room temperature. Samples were then washed in 1X PBS three times for 10 min. Oviduct muscle labeled with GFP was viewed with confocal microscopy. It was not reliable to measure muscle sarcomere length in the lateral oviduct. Therefore, our focal area was the upper common oviduct (Fig. 5g). Common oviduct sarcomere lengths were measured as distances between the centers of GFP bands. At least three sarcomeres were measured for each myofibril, and three myofibrils were averaged for each female reproductive tract. To avoid bias, the investigator was blinded to the origin of the reproductive tracts while measuring the muscle sarcomere length.

## Quantitative real-time PCR

Total RNA was extracted from different tissues (e.g., brain, VNC, and whole body of fly females) using the RNeasy Mini Kit (Qiagen, Cat# 74104) and then reverse transcribed into cDNA using HiScript III RT SuperMix for qPCR (Vazyme, Cat# R223-01) according to the manufacturer's protocol. qRT-PCR was performed in the AriaMx real-time PCR system (Agilent Technologies) with the ChamQ SYBR qPCR Master Mix Kit (Vazyme, Cat# Q311-02). Reactions were carried out for 30 s at 95 °C, followed by 45 cycles of three-step PCR for 10 s at 95 °C, 20 s at 55 °C, and 20 s at 72 °C. The RNA levels of the target genes were normalized to that of *tubulin* mRNA, and the relative concentration was determined using the $2^{-\Delta\Delta Ct}$ method. All the primers used for qRT-PCR in this study are listed in Supplementary Table 1.

## TRPA1 activation

Similar to the oviposition assay described above, *D. melanogaster* adults of *UAS-TRPA1*, *Tdc2-GAL4*, and *Tdc2-GAL4 > UAS-TRPA1* cohabited with Lb female wasps (exposed) or without any wasps (unexposed) at 23 °C (inactivating temperature) and 29 °C (activating temperature). The food dishes were replaced daily, and the eggs on the dishes were counted on Day 2.

## OA injection

OA (Sigma, Cat# 68631) was dissolved in ddH$_2$O at a final concentration of 100 μM. Approximately 20 nl of OA solution was injected into the abdomen of 3-day-old *Drosophila* female adults using the Eppendorf FemtoJet 4i device with the following parameters: injection pressure = 900 hPa; injection time = 0.15 s. After injection, the *D. melanogaster* adults cohabited with Lb female wasps (exposed) or without any wasps (unexposed) and were allowed to lay eggs for 1 day. The eggs on the dishes were counted.

## Statistics

All statistical analyses were performed in GraphPad Prism version 8.0 (GraphPad Software) and SPSS 26 (IBM). Normal distribution of the data was tested using the Shapiro−Wilk test. Bartlett chi-square test was used to test the homogeneity of variance of the data, which was consistent with the normal distribution. We used two-tailed unpaired Student's t tests and Mann-Whitney U test to determine the statistical significance of a difference between two treatments. ANOVA with Sidak's multiple comparisons tests and Kruskal−Wallis test with Dunn's multiple comparisons test were used to compare mean differences between multiple groups. Details of the statistical analysis are provided in the figure legends, including how significance was defined and the statistical methods used, and raw statistics data were provided in Source Data file. Data represent the mean ± standard error of the mean (SEM). Different letters in Fig. 2a and Fig. 3a, c, d indicate statistically significant differences ($p < 0.05$). For all other tests, significance values are indicated as *$p < 0.05$; **$p < 0.01$; ***$p < 0.001$.

## Reporting summary

Further information on research design is available in the Nature Research Reporting Summary linked to this article.

## Data availability

All data supporting the findings of this study are available within the paper and its supplementary information files. Source data are provided with this paper.

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

## Acknowledgements

We thank Drs. Shuai Zhan and Yong Zhang for critically reading the manuscript. We thank Bloomington Drosophila Stock Center, Tsing Hua Fly Center, Drs. Dan Hultmark, István Andó, Zefeng Gong, Yi Sun and Jia Huang for providing parasitoid wasps and *Drosophila* stocks. We also thank Dr. Zhangwu Zhao for providing NPF antibody. This study was supported by the Zhejiang Provincial Natural Science Foundation of China (LR18C140001) to J.H., the National Science Foundation of China (Grant Nos. 32172467 and 31622048) to J.H., the Key International Joint Research Program of National Natural Science Foundation of China (31920103005) to X.C., and the Fundamental Research Funds for the Central Universities (2021FZZX001-31) to J.H. and X.C.

## Author contributions

J.H. conceived the project. J.H. designed and directed the studies. L.P., Z.L., J.C., Z.D., and S.Z. performed and analysed the egg-laying assays and search behavior experiments. L.P. and Q.Z. performed qRT-PCR experiments. L.P., Y. L., and Y.S. performed the immunohistochemistry experiments. L.P., X.C., and J.H. interpreted the data. L.P. and J.H. wrote the manuscript. All authors have read and approved the manuscript submission.

## Competing interests

The authors declare no competing interests.
