## [Peer Review File · Nature Communications]

Search performance and octopamine neuronal signaling mediate parasitoid induced changes in *Drosophila* oviposition behaviorREVIEWER COMMENTS

Reviewer #1 (Remarks to the Author):

A fundamental question in neuroscience and ecology is how animals adaptively modify their behaviors in response to an ever-changing environment including threats from predators/parasites. In particular, in an environment where threats from predators/parasites are inevitable, e.g., 90% of natural populations of *Drosophila melanogaster* have been attacked by parasitoid wasps, it is of particular significance for the host flies to adjust behaviors by evaluating the level of risks from potential threats. In this manuscript, Pang et al., found that *Drosophila* females adjust egg-laying behaviors by assessing threats from parasitoid wasps such as *Leptopilina boulardi* (Lb). Previous studies have found that presence of Lb females, which lay eggs inside the *Drosophila* larvae, elicited defensive egg-laying behaviors in *Drosophila* females. Pang et al., found that *Drosophila* females showed different egg-laying behaviors to young and aged Lb females, which have different levels of threats (lower threat from aged Lb females). They further found that such an adaptive defense depends mainly on visual assessment of locomotor behaviors of Lb females through a characterized subset of LC4 interneurons in the *Drosophila* visual system. They also found that exposure to young Lb females, but not aged Lb females, induced downregulation of the octopamine (OA) signaling, as well as decreased follicle rupture and Mmp2 activity, and therefore reduced ovulation in flies. The strength of this manuscript is to utilize genetic manipulations in both the parasitoid wasp Lb and host flies, and illustrate a mechanism from sensing different levels of threats to adaptive changes in a molecular pathway and corresponding behaviors, which is significant in both neuroscience and ecology.

A major point:

In the first figure, the authors nicely showed that *Drosophila* females decreased egg-laying behaviors in response to young but not aged Lb female wasps, and conclude that such adaptation is not due to habituation, e.g., days long housing with Lb females. While the authors did show that female flies exposed to young Lb females still reduced egg-laying after a week-long housing with Lb females, it does not preclude the possibility of housing experience with Lb females may still affect flies' oviposition behavior. Authors are encouraged to test naïve female flies housed with aged Lb females to check if females flies, without any prior experience with Lb females, would decreased egg-laying. If they do, is the level of reduction similar to flies exposed to young Lb females? This is possible as behavioral adaption certainly also involved mechanisms of learning through experiences.

Minor points:

1. In figure 2, authors found that the effect of egg laying providing only visual stimulus of Lb females is weaker than that in regular fly bottles, and proposed that such difference may be due to *Drosophila* females not seeing Lb females clearly in the visual-based apparatus. I think it is also possible that other sensory modalities, such as olfaction and audition, would strength the visual-induced egg laying reduction, while olfaction and audition are not sufficient, without vision, to affect egg-laying behavior.
2. The host search performance of Lb wasps is not well defined in the methods section. Authors should describe how they analyze the search performance index.
3. Authors should provide more detailed information, e.g., the cat# of antibodies used in the methods section.

Overall, this is an interesting and exciting manuscript that tackles the challenging scientific question in neuroscience and ecology how animals evaluate different levels of threats and adaptively modify their behaviors, and deserves publication in *Nature Communications* upon some revisions as listed above.

Reviewer #2 (Remarks to the Author):

In this manuscript, “Octopamine neuronal signaling mediates predator-induced changes in *Drosophila* oviposition behavior”, Pang et al. report a mechanistic pathway in which the presence of young parasitoid wasps alters *Drosophila* egg laying behavior, but older wasps have no such effect. The authors show that a set of visual neurons, LC4, detect the locomotor activity of young, active wasps. Further, they demonstrated that octopaminergic neurons in the ventral nerve cord downregulate octopamine when wasps are present leading to egg retention in the ovaries, and that this likely contributes to the reduced oviposition phenotype.

There are many new and interesting findings in this manuscript, opening new avenues to pursue the mechanistic basis of fly anti-parasitoid behaviors, and defensive behaviors in general. The finding that old wasps do not elicit any oviposition reduction behavior, likely due to reduced host searching behavior, is novel. Furthermore, other studies have shown that small neuropeptides regulate fly behaviors post-wasp exposure; the finding here about the neuromodulator octopamine expands the list of important signaling molecules for fly behavioral defense responses.

This is an easy-to-read paper with clear figures. Below are some points that the authors might consider to enhance the manuscript.

Major points:

1. Are the aged wasps given any hosts to parasitize before the experiments start? This is unclear in the methods and needs to be addressed. It is my understanding that these wasps are pro-ovigenic, meaning they have produced the full complement of mature eggs at eclosion. This means that once the wasps are depleted of eggs they may stop showing infection behavior and their risk to fly offspring is canceled. Or perhaps older wasps resorb unused eggs over time? I think it's important to know whether the aged wasps have a reduced complement of eggs to lay, or are seen as less of a threat by the flies for other reasons. Figure 3 shows that older wasps move and search less – is this connected to a lack of eggs available to lay? This question should be addressed by additional experiments or at least by extra discussion.
2. I'm curious as to how the authors think the LC4 visual neurons regulate octopaminergic signaling. Previous studies have shown that LC4 is activated by a looming stimulus – specifically the velocity of the looming stimulus. In turn, these cells synapse onto the giant fiber neurons to induce a quick jump response (Wu et al., 2016, eLife; von Reyn et al., 2017, Neuron; Ache et al., 2019, Curr Biol). How do the authors presume that these LC4 cells contribute to both wasp-induced oviposition reduction and looming stimulus-induced escape response? Do the wasps ‘loom’? Additional experiments or at least some extra discussion is required.

Minor points:

1. Although other authors have used the term “predator” to describe these wasps, the more accurate terminology is “parasitoid”.
2. Given intraspecific variation in wasp characteristics, please provide the strain names and/or provenance of the wasp strains used in this study.
3. Line 87: “monitored egg laying for a much longer time”. Longer compared to what?
4. Is the aging effect common across all wasps or specific to Lb? Supplementary Figure 1 shows other *Drosophila* wasps, Lh and Aj, but old females were not tested.
5. Line 120: please change to “2nd instar larvae”
6. Line 142-146: “To further confirm that *Drosophila* females can sense the distinct life-threatening stimuli of the cohabiting Lb female wasps, we generated two olfaction-defective strains, including a strain with knockdown of Orco (a gene encoding an obligate coreceptor of all odorant receptor proteins) mediated by RNA interference (Supplementary Fig. 2b) and an antenna-ablated strain.” This is confusing as it implies that the flies rather than the wasps are olfaction-deficient. Please consider revising.
7. The authors show that the olfaction deficient mutant Orco[1] acts like wild type flies in the presence of wasps (ie reduces egg laying, Figure 2b). There is a contradiction in the literature about whether

olfaction is required, and it seems that studies using Orco[1] show that olfaction is not required while studies using Orco[2] find that olfaction is required. It might be worth confirming the current results that olfaction is not required using a second fly olfaction mutant.

8. How are the flies distinguishing young from old wasps in the apparatus shown in Figure 2E? Given there are no fly larvae to infect on the wasp side, I assume neither set of wasps is showing oviposition behavior?

9. Line 180: "We found that parental control flies expressing a UAS-TNT transgene alone..." If this element is alone, it should not be expressing anything. Please clarify.

10. Line 193-194: "because it is not possible for flies to directly observe the parasitic rate of Lb". I'm not sure what is meant here. Why can't flies observe wasps infecting fly larvae?

11. Figure 3b (wasp locomotor trajectories) is confusing. To me, they all look very similar. What is the evidence that "young females moved more on the food medium than did the old wasps..." (lines 202-203)? Does the analysis consist of the number of green lines, the distance wasps walked, etc?

12. There are a few instances where the figure legend provides the replicate values for specific experiments, but the number of n values does not match the number of samples in the figures. For example, figure 3c (line 517) provides six n values but there are only five groups in the figure. This occurs again in figure 6c. Please correct.

13. Line 344-345: "showed a significant increase in oviposition rate relative to the controls regardless of whether they were exposed to young Lb females (Fig. 6e)". To me it looks like the third green (unexposed) bar is the same height as the other green bars, and it is only the exposed flies that show increased egg laying. Am I missing something?

14. The authors have successfully activated Tdc2 neurons by using the thermogenetic TrpA1 channel. I'm curious if activation of the visual LC4 neurons could also induce the oviposition reduction behavior in the absence of wasps. This would demonstrate that these visual neurons are sufficient, in addition to being necessary, for wasp detection.

Reviewer #3 (Remarks to the Author):

In the manuscript by Pang et al, a variety of manipulations are used to demonstrate that *Drosophila* females have distinct egg-laying response when exposed to young and old wasps through visual system. The evidence presented by the authors suggests that flies rely on LC4 visual projection neuron to sense young and old female wasps. When flies see young female wasps, the expression of Tdc2 and Tβh were significantly decreased in VNC, leads to reduce the level of octopamine, which in turn impairs mature follicle cell rupture, resulting in ovulation defects. Although the molecular mechanism is quite well documented, the most amazing behavior assays attract the egg ball the most and require to pay more attention. I would like recommend it for publication in NC after several relatively small issues are addressed.

1) The most important novelty in the manuscript is the behavior assay rather than the molecular mechanism. In the title, abstract, results, and discussion, the behavior studies should be stressed in a great deal.

2) Figure 1. It will be much better to perform more and solid experiments to determine the young wasp has a higher impact on *Drosophila* oviposition than the old wasp, including direct comparison of young and old wasps exposed to young wasps, and a shift from old to young wasp exposed to young wasps.

3) Results 2 should be separated using two sub-titles. The predation efficiency is very important and should put into a major figure.

4) The wasp studies are much more attractive than the genetic and molecular aspects in *Drosophila*. In the context, it mixes wasp studies with *Drosophila* studies very often. Fig. 2 and Fig. 3 should be converted so that the audience is able to easily follow the authors' logic. In this way, the audiences can shift their attention from wasp behaviors to *Drosophila* in general.

5) Oviposition rate decreases were not observed when flies cohabited in darkness with either young or old wasps in Fig2C. Why? A visually impaired mutant *ninaB* could be used from additional evidence of the visual cues.

6) OA signaling plays a crucial role in a variety of behaviors, including the fight-or-flight response,

motivation, aggression, and reproduction. the authors used qPCR to examine the expression levels of Tdc2 and T β h. One would expect that anti-Tcd2 should be present in brain and VNC. The OA receptor could be detected to strength of the manuscript.

7) Discussion. Mating response and hormone regulation are totally neglected, and important references are not included.

Reviewer #1 (Remarks to the Author):

A fundamental question in neuroscience and ecology is how animals adaptively modify their behaviors in response to an ever-changing environment including threats from predators/parasites. In particular, in an environment where threats from predators/parasites are inevitable, e.g., 90% of natural populations of *Drosophila melanogaster* have been attacked by parasitoid wasps, it is of particular significance for the host flies to adjust behaviors by evaluating the level of risks from potential threats. In this manuscript, Pang et al., found that *Drosophila* females adjust egg-laying behaviors by assessing threats from parasitoid wasps such as *Leptopilina bouleari* (Lb). Previous studies have found that presence of Lb females, which lay eggs inside the *Drosophila* larvae, elicited defensive egg-laying behaviors in *Drosophila* females. Pang et al., found that *Drosophila* females showed different egg-laying behaviors to young and aged Lb females, which have different levels of threats (lower threat from aged Lb females). They further found that such an adaptive defense depends mainly on visual assessment of locomotor behaviors of Lb females through a characterized subset of LC4 interneurons in the *Drosophila* visual system. They also found that exposure to young Lb females, but not aged Lb females, induced downregulation of the octopamine (OA) signaling, as well as decreased follicle rupture and Mmp2 activity, and therefore reduced ovulation in flies. The strength of this manuscript is to utilize genetic manipulations in both the parasitoid wasp Lb and host flies, and illustrate a mechanism from sensing different levels of threats to adaptive changes in a molecular pathway and corresponding behaviors, which is significant in both neuroscience and ecology.

AUTHORS' RESPONSE: We appreciate your high evaluation on our study and valuable suggestions that help improve the manuscript.

A major point:

In the first figure, the authors nicely showed that *Drosophila* females decreased egg-laying behaviors in response to young but not aged Lb female wasps, and conclude that such adaptation is not due to habituation, e.g., days long housing with Lb females. While the authors did show that female flies exposed to young Lb females still reduced egg-laying after a week-long housing with Lb females, it does not preclude the possibility of housing experience with Lb females may still affect flies' oviposition behavior. Authors are encouraged to test naïve female flies housed with aged Lb females to check if females flies, without any prior experience with Lb females, would decreased egg-laying. If they do, is the level of reduction similar to flies exposed to young Lb females? This is possible as behavioral adaptation certainly also involved mechanisms of learning through experiences.

AUTHORS' RESPONSE: This is an important point. We agree with your concern that additional experiments are needed to preclude the possibility of housing experience with Lb females that may contribute to the oviposition defense behavior. As per your suggestion, we have tested the naïve female flies

housed with different aged Lb females directly to check the level of oviposition reduction. Interestingly, we found that the female flies, without any prior experience with Lb females, showed the similar level of decreased egg-laying to flies that were continuously exposed to young Lb. We have mentioned these results in both main text (lines 117-124) and a new supplementary Fig. 1.

Minor points:

1. In figure 2, authors found that the effect of egg laying providing only visual stimulus of Lb females is weaker than that in regular fly bottles, and proposed that such difference may be due to *Drosophila* females not seeing Lb females clearly in the visual-based apparatus. I think it is also possible that other sensory modalities, such as olfaction and audition, would strength the visual-induced egg laying reduction, while olfaction and audition are not sufficient, without vision, to affect egg-laying behavior.

AUTHORS' RESPONSE: Thank you. We agree with your point that it is possible that some other sensory modalities, e.g., olfaction and audition, might strength the visual-induced egg-laying reduction, while they are not sufficient alone. This extra explanation has been presented in the main text (see lines 229-230 of our revised manuscript).

2. The host search performance of Lb wasps is not well defined in the methods section. Authors should describe how they analyze the search performance index.

AUTHORS' RESPONSE: We apologize for the incomplete information of defining the host search performance in Methods section. We have extended the Methods section with details to address your concerns (see lines 742-744 for host search performance, lines 745-748 for the analysis of search index).

3. Authors should provide more detailed information, e.g., the cat# of antibodies used in the methods section.

AUTHORS' RESPONSE: We apologize for the incomplete information of the Cat# of the antibodies, kits, and chemicals that we used in this study. As per your suggestion, we have added these details in the Methods section (see lines 770, 783, 786, 788, 832, 856, 857, 859 and 872).

Overall, this is an interesting and exciting manuscript that tackles the challenging scientific question in neuroscience and ecology how animals evaluate different levels of threats and adaptively modify their behaviors, and deserves publication in Nature Communications upon some revisions as listed above.

AUTHORS' RESPONSE: Thank you. Your comments and kindly suggestions help improve the manuscript greatly.

Reviewer #2 (Remarks to the Author):

In this manuscript, “Octopamine neuronal signaling mediates predator-induced changes in *Drosophila* oviposition behavior”, Pang et al. report a mechanistic pathway in which the presence of young parasitoid wasps alters *Drosophila* egg laying behavior, but older wasps have no such effect. The authors show that a set of visual neurons, LC4, detect the locomotor activity of young, active wasps. Further, they demonstrated that octopaminergic neurons in the ventral nerve cord downregulate octopamine when wasps are present leading to egg retention in the ovaries, and that this likely contributes to the reduced oviposition phenotype.

There are many new and interesting findings in this manuscript, opening new avenues to pursue the mechanistic basis of fly anti-parasitoid behaviors, and defensive behaviors in general. The finding that old wasps do not elicit any oviposition reduction behavior, likely due to reduced host searching behavior, is novel.

Furthermore, other studies have shown that small neuropeptides regulate fly behaviors post-wasp exposure; the finding here about the neuromodulator octopamine expands the list of important signaling molecules for fly behavioral defense responses.

This is an easy-to-read paper with clear figures. Below are some points that the authors might consider to enhance the manuscript.

AUTHORS' RESPONSE: We thank the reviewer for being interested in our work and for the valuable suggestions on how to improve it.

Major points:

1. Are the aged wasps given any hosts to parasitize before the experiments start? This is unclear in the methods and needs to be addressed. It is my understanding that these wasps are pro-ovigenic, meaning they have produced the full complement of mature eggs at eclosion. This means that once the wasps are depleted of eggs they may stop showing infection behavior and their risk to fly offspring is canceled. Or perhaps older wasps resorb unused eggs over time? I think it's important to know whether the aged wasps have a reduced complement of eggs to lay, or are seen as less of a threat by the flies for other reasons. Figure 3 shows that older wasps move and search less – is this connected to a lack of eggs available to lay? This question should be addressed by additional experiments or at least by extra discussion.

AUTHORS' RESPONSE: All the wasps were not given any hosts to parasitize before starting the experiments. We apologize for the incomplete information in the original manuscript. This Method information has been extended in the revised manuscript (see lines 704-705).

It is true that the Lb wasps are pro-ovigenic. We understand your concerns that it is possible that the aged wasps have a reduced complement of eggs which may be connected to the impaired host search performance, representing the less of a

threat to the flies. To address your concerns, we have newly added two aspects of work. First, we have dissected and taken the ovary images of young (4-day-old) and old (12-day-old) female Lb wasps. The results showed a comparable size (newly added Supplementary Fig. 3a). Furthermore, we have quantified the difference by counting all the mature eggs of young (4-day-old) and old (12-day-old) female Lb wasps. No significant difference was observed (newly added Supplementary Fig. 3b). We have also mentioned these results in the main text (lines 183-187) of the revised manuscript.

2. I'm curious as to how the authors think the LC4 visual neurons regulate octopaminergic signaling. Previous studies have shown that LC4 is activated by a looming stimulus – specifically the velocity of the looming stimulus. In turn, these cells synapse onto the giant fiber neurons to induce a quick jump response (Wu et al., 2016, eLife; von Reyn et al., 2017, Neuron; Ache et al., 2019, Curr Biol). How do the authors presume that these LC4 cells contribute to both wasp-induced oviposition reduction and looming stimulus-induced escape response? Do the wasps 'loom'? Additional experiments or at least some extra discussion is required.

AUTHORS' RESPONSE: This is another important point. It is very interesting that the LC4 neurons contribute to both wasp-induced oviposition reduction and looming stimulus-induced escape response. Our results have shown that the reduction of egg laying in the transparent chamber is much weaker than that in regular fly bottles, potentially reflecting the different effects between far and close visual stimulus (Fig. 4f). As such, it seems possible that the parasitic wasps may present some certain of looming stimulus to flies. In addition, the functional relevance of the different visual cues in LC4 cells and their downstream signals remains to be determined. Thank you for the suggestions, we have added a new paragraph in Discussion section (lines 435-445).

Minor points:

1. Although other authors have used the term “predator” to describe these wasps, the more accurate terminology is “parasitoid”.

AUTHORS' RESPONSE: Thank you. We agree with your point that “parasitoid” is more accurate than “predator” to describe the wasps. It has been corrected in the revised manuscript. In addition, we have added a sentence “Parasitoids are deadly natural enemies representing a special kind of predators who usually present symbiotic relationship with their prey animals (known as hosts) in the wild” in the main text (lines 60-62) to make the transition frequently from “predator” to “parasitoid” in the Introduction section.

2. Given intraspecific variation in wasp characteristics, please provide the strain names and/or provenance of the wasp strains used in this study.

AUTHORS' RESPONSE: We have provided the details of wasp characteristics

in the Method section (lines 692-694).

3. Line 87: “monitored egg laying for a much longer time”. Longer compared to what?

AUTHORS’ RESPONSE: We have changed this sentence as “monitored egg laying for a much long time of approximately 20 days” (lines 88-89).

4. Is the aging effect common across all wasps or specific to Lb? Supplementary Figure 1 shows other *Drosophila* wasps, Lh and Aj, but old females were not tested.

AUTHORS’ RESPONSE: Thank you. To address your concerns, we have additionally monitored the egg-laying numbers for 12 days in the oviposition assay. The results have been mentioned in both main text (lines 133-136) and new Supplementary Figs. 2a and 2b. We note that fly females laid fewer eggs during approximately the first 8 days in the presence of Lh or Aj female wasps, and the effects were similar to those following exposure to young female Lb (Supplementary Figs. 1a and 1b). However, when flies were exposed to old Lh or Aj wasps, oviposition numbers were equivalent to that of the unexposed flies. The results indicate that the aging effect is common across the *Drosophila* wasps.

5. Line 120: please change to “2nd instar larvae”

AUTHORS’ RESPONSE: It has been changed as suggested (see line 130).

6. Line 142-146: “To further confirm that *Drosophila* females can sense the distinct life-threatening stimuli of the cohabiting Lb female wasps, we generated two olfaction-defective strains, including a strain with knockdown of *Orco* (a gene encoding an obligate coreceptor of all odorant receptor proteins) mediated by RNA interference (Supplementary Fig. 2b) and an antenna-ablated strain.” This is confusing as it implies that the flies rather than the wasps are olfaction-deficient. Please consider revising.

AUTHORS’ RESPONSE: To avoid confusion, we have modified this sentence as “To further identify the distinct life-threatening stimuli from the cohabiting Lb females, we generated two olfaction-defective wasp strains, including a strain with knockdown of *Orco* (a gene encoding an obligate coreceptor of all odorant receptor proteins) mediated by RNA interference (Fig. 2b) and an antenna-ablated strain” (see lines 155-159).

7. The authors show that the olfaction deficient mutant *Orco*[1] acts like wild type flies in the presence of wasps (ie reduces egg laying, Figure 2b). There is a contradiction in the literature about whether olfaction is required, and it seems that studies using *Orco*[1] show that olfaction is not required while studies using *Orco*[2]

find that olfaction is required. It might be worth confirming the current results that olfaction is not required using a second fly olfaction mutant.

AUTHORS' RESPONSE: As per your suggestion, we have additionally tested another olfaction deficient mutant *Orco*² in the oviposition assay. Similar to *Orco*¹, the *Orco*² flies initially had reduced oviposition rates in the presence of female wasps, but rates gradually returned to normal, confirming that olfaction is not required. We have mentioned these results in the revised manuscript (lines 213-216) and a new Supplementary Fig. 4a.

8. How are the flies distinguishing young from old wasps in the apparatus shown in Figure 2E? Given there are no fly larvae to infect on the wasp side, I assume neither set of wasps is showing oviposition behavior?

AUTHORS' RESPONSE: Thank you. We apologize for the incomplete information in the original manuscript that might cause your confusion. Actually, the *Leptopilina* parasitoids normally show host search behavior, no matter whether the fly larvae exist or not (also see Supplementary Video 1). We have mentioned this characteristic in Method section of revised manuscript (lines 744-745).

9. Line 180: "We found that parental control flies expressing a UAS-TNT transgene alone..." If this element is alone, it should not be expressing anything. Please clarify.

AUTHORS' RESPONSE: It is true that if the element is alone, it should express nothing. We have changed this sentence as "We found that parental control flies containing a UAS-TNT transgene alone..." (line 237).

10. Line 193-194: "because it is not possible for flies to directly observe the parasitic rate of Lb". I'm not sure what is meant here. Why can't flies observe wasps infecting fly larvae?

AUTHORS' RESPONSE: The flies can observe wasps infecting host larvae. We meant to express that the flies can't directly get the results of parasitic rate. We apologize for the confusion, and this sentence has been changed as "because it is not possible for flies to directly obtain the parasitic rate of Lb" (line 171).

11. Figure 3b (wasp locomotor trajectories) is confusing. To me, they all look very similar. What is the evidence that "young females moved more on the food medium than did the old wasps..." (lines 202-203)? Does the analysis consist of the number of green lines, the distance wasps walked, etc?

AUTHORS' RESPONSE: Thank you. The green lines represent the distance that wasps walked in the locomotion trajectory assay. We have added a new Fig. 3c to

show the walk distance for each type of parasitic wasps. It is clear now that “young females moved more on the food medium than did the old wasps...”. We have mentioned this analysis in the main text (lines 180-181 and 197-199). We hope these revisions might make this section less confused.

12. There are a few instances where the figure legend provides the replicate values for specific experiments, but the number of n values does not match the number of samples in the figures. For example, figure 3c (line 517) provides six n values but there are only five groups in the figure. This occurs again in figure 6c. Please correct.

AUTHORS' RESPONSE: Thank you. We apologize for the inconsistent n values in the Figs. 3c and 6c and in their legends of the original manuscript. They have been corrected as suggested (lines 551 and 647).

13. Line 344-345: “showed a significant increase in oviposition rate relative to the controls regardless of whether they were exposed to young Lb females (Fig. 6e)”. To me it looks like the third green (unexposed) bar is the same height as the other green bars, and it is only the exposed flies that show increased egg laying. Am I missing something?

AUTHORS' RESPONSE: Thank you. We apologize for the inappropriate writings of results regarding this Fig. 6e in the original manuscript. Actually, we meant to express that there is a significant increase in oviposition rate relative to the two exposed controls (red bars). We have changed this sentence as “showed a significant increase in oviposition rate relative to the controls that were exposed to young Lb females” (lines 365-366).

14. The authors have successfully activated Tdc2 neurons by using the thermogenetic TrpA1 channel. I'm curious if activation of the visual LC4 neurons could also induce the oviposition reduction behavior in the absence of wasps. This would demonstrate that these visual neurons are sufficient, in addition to being necessary, for wasp detection.

AUTHORS' RESPONSE: Thank you. As per your suggestion, we have performed an extra experiment to activate LC4 neurons by using the thermogenetic TrpA1 channel. Unfortunately, we failed to get the oviposition reduction behavior (see below).

We have modified our writings as “these neurons are necessary to initiate the effect of young female wasps” in the revised manuscript (lines 240-241), and we believe that the above new findings support our conclusion that these LC4 neurons are necessary but not sufficient for wasp detection.

Reviewer #3 (Remarks to the Author):

In the manuscript by Pang et al, a variety of manipulations are used to demonstrate that *Drosophila* females have distinct egg-laying response when exposed to young and old wasps through visual system. The evidence presented by the authors suggests that flies rely on LC4 visual projection neuron to sense young and old female wasps. When flies see young female wasps, the expression of Tdc2 and Tβh were significantly decreased in VNC, leads to reduce the level of octopamine, which in turn impairs mature follicle cell rupture, resulting in ovulation defects. Although the molecular mechanism is quite well documented, the most amazing behavior assays attract the egg ball the most and require to pay more attention. I would like recommend it for publication in NC after several relatively small issues are addressed.

AUTHORS' RESPONSE: We appreciate your high evaluation and kindly suggestions on our study.

1) The most important novelty in the manuscript is the behavior assay rather than the molecular mechanism. In the title, abstract, results, and discussion, the behavior studies should be stressed in a great deal.

AUTHORS' RESPONSE: Thank you. As per your suggestion, we have changed the title to “The search performance and octopamine neuronal signaling mediate parasitoid-induced changes in *Drosophila* oviposition behavior”. We have also expanded and reorganized the text to emphasize the results from behavior assay in other sections including abstract, results and discussion (see lines 26-28, 126-206, 435-445).

2) Figure 1. It will be much better to perform more and solid experiments to determine the young wasp has a higher impact on *Drosophila* oviposition than the old wasp, including direct comparison of young and old wasps exposed to young wasps, and a shift from old to young wasp exposed to young wasps.

AUTHORS' RESPONSE: This is an important point. We agree with your concern that additional experiments are needed to determine the young wasps have a higher impact on *Drosophila* oviposition than the old wasps. Although the shifting experiments in our original manuscript have shown that the young wasps do affect the oviposition (Fig. 1c, old to young; Fig. 1d, old to old), we have performed extra experiments to compare the oviposition rate. When the female flies were directly housed with different aged Lb females (a strategy for direct comparison), they showed a decreased egg laying only in the presence of young wasps but not the old wasps. We have mentioned these results in both main text (lines 117-123) and a new supplementary Fig. 1 of the revised manuscript.

3) Results 2 should be separated using two sub-titles. The predation efficiency is very important and should put into a major figure.

AUTHORS' RESPONSE: Thank you. We agree that the parasitic efficiency is very important. We have separated Results 2 using two sub-titles, and also moved the Supplementary Fig. 2 in the original manuscript to the main Fig. 2 (lines 126-166).

4) The wasp studies are much more attractive than the genetic and molecular aspects in *Drosophila*. In the context, it mixes wasp studies with *Drosophila* studies very often. Fig. 2 and Fig. 3 should be converted so that the audience is able to easily follow the authors' logic. In this way, the audiences can shift their attention from wasp behaviors to *Drosophila* in general.

AUTHORS' RESPONSE: Thank you. As per your suggestion, we have converted the two parts to make it shift smoothly (lines 168-206 and lines 208-245).

5) Oviposition rate decreases were not observed when flies cohabited in darkness with either young or old wasps in Fig2C. Why? A visually impaired mutant *ninaB* could be used from additional evidence of the visual cues.

AUTHORS' RESPONSE: We apologize for the unclear sentences. We have expanded the conclusion of the results as "Oviposition rate decreases were not observed when the flies cohabited in darkness with either young (4-day-old) or old (12-day-old) female parasitoid wasps, confirming the findings that vision is very important to the decreased egg laying" in the revised manuscript (lines 221-222). We also thank the reviewer for encouraging us to provide additional evidence of the visual cues. We have additionally tested the visually impaired mutant, *ninaB*¹, and found they showed equivalent egg laying in the presence and absence of Lb females. These results further confirmed our conclusion that the vision is responsible for decreased oviposition. We have mentioned these results in both main text (lines 216-219) and a new Supplementary Fig. 4b.

6) OA signaling plays a crucial role in a variety of behaviors, including the fight-or-flight response, motivation, aggression, and reproduction. The authors used qPCR to examine the expression levels of *Tdc2* and *Tβh*. One would expect that anti-*Tcd2* should be present in brain and VNC. The OA receptor could be detected to strength of the manuscript.

AUTHORS' RESPONSE: We agree with the reviewer that *Tdc2* is specifically expressed in the brain and VNC. That's why we analyzed the expression of the mRNAs encoding *Tdc2* and *Tβh* in the brain and VNC of the exposed and unexposed *Drosophila* females by qRT-PCR. Because OA signaling plays a crucial role in a variety of behaviors, one would expect that the OA receptor is

widely presented in multiple tissues. As per your suggestion, we have additionally tested the expression levels of OA receptors (e.g., Octa2R, Octβ1R, Octβ2R, Octβ3R and Oct-TyrR) in the whole bodies of flies after exposure to young Lb wasps on Day 2. The results (see below) showed that there was no difference of tested OA receptor genes between the exposed and unexposed flies. These results did not change our conclusion that the levels of OA were significantly decreased in VNC after exposure to young female wasps.

7) Discussion. Mating response and hormone regulation are totally neglected, and important references are not included.

AUTHORS' RESPONSE: Thank you for reminding us of the function of mating and hormone regulation in charge of insect egg laying. We have added some important references in Discussion section of the revised manuscript (lines 468-469).

REVIEWER COMMENTS

Reviewer #1 (Remarks to the Author):

The authors have done an incredible job to revise the manuscript. In particular, they have provided substantial behavioral experiments to test the potential role of prior social experiences in the modification of egg-laying behaviors. They also used additional vision or olfaction mutants to support their conclusions. I agree with the Reviewer 3 that the wasp-related behavioral assays are the most attractive parts of the manuscript, and the corresponding textual changes in the revised manuscript are appropriate. I have no further concern and support the publication of the manuscript in its current form.

Reviewer #2 (Remarks to the Author):

In this revised version of the manuscript, “Octopamine neuronal signaling mediates predator-induced changes in *Drosophila* oviposition behavior”, Pang et al. have thoroughly responded to the critiques of all three reviewers and perform a variety of new experiments to strengthen the manuscript. I have a few remaining minor suggestions:

- (1) I appreciate the reason that the title was changed but I think the current title will be confusing to readers. Maybe change to something like “Visual recognition of parasitoid searching behaviors mediates a defensive oviposition behavior in *Drosophila*”.
- (2) Line 27: Consider changing the new sentence in the abstract to something like “This effect is controlled by differences in the search performance of young and old female wasps.”
- (3) Line 88: Consider changing the new sentence in the abstract to something like “for a much longer time than in previous experiments – approximately 20 days.”
- (4) Line 165: Consider changing the sentence to something like “suggest that wasp parasitism behaviors might cause the decreased oviposition of flies exposed to young parasitoids.”
- (5) Line 243: I am curious why the authors chose to omit their new data showing that TrpA1 activation of LC4 neurons does not induce the egg reduction behavior? It seems interesting to show that Lc4 activation is necessary but not sufficient.

Reviewer #3 (Remarks to the Author):

The authors have addressed all my concerns. The manuscript has been significantly improved and is ready for publication in Nature Communications.

Sheng Li

REVIEWERS' COMMENTS

Reviewer #1 (Remarks to the Author):

The authors have done an incredible job to revise the manuscript. In particular, they have provided substantial behavioral experiments to test the potential role of prior social experiences in the modification of egg-laying behaviors. They also used additional vision or olfaction mutants to support their conclusions. I agree with the Reviewer 3 that the wasp-related behavioral assays are the most attractive parts of the manuscript, and the corresponding textual changes in the revised manuscript are appropriate. I have no further concern and support the publication of the manuscript in its current form.

AUTHORS' RESPONSE: We appreciate your high evaluation and inputs that help improve the study.

Reviewer #2 (Remarks to the Author):

In this revised version of the manuscript, “Octopamine neuronal signaling mediates predator-induced changes in *Drosophila* oviposition behavior”, Pang et al. have thoroughly responded to the critiques of all three reviewers and perform a variety of new experiments to strengthen the manuscript. I have a few remaining minor suggestions:

AUTHORS’ RESPONSE: We appreciate your high evaluation and inputs that help improve the study.

(1) I appreciate the reason that the title was changed but I think the current title will be confusing to readers. Maybe change to something like “Visual recognition of parasitoid searching behaviors mediates a defensive oviposition behavior in *Drosophila*”.

AUTHORS’ RESPONSE: Thank you. Based on the suggestions from both the Reviewer and Editor, we have revised our manuscript title as “Search performance and octopamine neuronal signaling mediate parasitoid induced changes in *Drosophila* oviposition behavior”.

(2) Line 27: Consider changing the new sentence in the abstract to something like “This effect is controlled by differences in the search performance of young and old female wasps.”

AUTHORS’ RESPONSE: Thank you. It has been corrected as suggested.

(3) Line 88: Consider changing the new sentence in the abstract to something like “for a much longer time than in previous experiments – approximately 20 days.”

AUTHORS’ RESPONSE: Thank you. It has been corrected as suggested (line 82).

(4) Line 165: Consider changing the sentence to something like “suggest that wasp parasitism behaviors might cause the decreased oviposition of flies exposed to young parasitoids.”

AUTHORS’ RESPONSE: Thank you. It has been corrected as suggested (line 159).

(5) Line 243: I am curious why the authors chose to omit their new data showing that TrpA1 activation of LC4 neurons does not induce the egg reduction behavior? It seems interesting to show that Lc4 activation is necessary but not sufficient.

AUTHORS’ RESPONSE: Thank you. We have provided the related data in a

new supplementary Fig. 5 in the revised manuscript, and also mentioned these results in the main text (lines 234-239).

Reviewer #3 (Remarks to the Author):

The authors have addressed all my concerns. The manuscript has been significantly improved and is ready for publication in Nature Communications.

AUTHORS' RESPONSE: We appreciate your high evaluation and kindly suggestions on our study.